# Antiviral action of a functionalized plastic surface against human coronaviruses

Sailee Shroff,[1] Marjo Haapakoski,[1] Kosti Tapio,[2,3] Mira Laajala,[1] Miika Leppänen,[1] Zlatka Plavec,[4,5] Antti Haapala,[6,7] Sarah J. Butcher,[4,5] Janne A. Ihalainen,[1] J. Jussi Toppari,[2] Varpu Marjomäki[1]

**ABSTRACT**  Viruses may persist on solid surfaces for long periods, which may contribute to indirect transmission. Thus, it is imperative to develop functionalized surfaces that will lower the infectious viral load in everyday life. Here, we have tested a plastic surface functionalized with tall oil rosin against the seasonal human coronavirus OC43 as well as severe acute respiratory syndrome coronavirus 2. All tested non-functionalized plastic surfaces showed virus persistence up to 48 h. In contrast, the functionalized plastic showed good antiviral action already within 15 min of contact and excellent efficacy after 30 min over 90% humidity. Excellent antiviral effects were also observed at lower humidities of 20% and 40%. Despite the hydrophilic nature of the functionalized plastic, viruses did not adhere strongly to it. According to helium ion microscopy, viruses appeared flatter on the rosin-functionalized surface, but after flushing away from the rosin-functionalized surface, they showed no apparent structural changes when imaged by transmission electron microscopy of cryogenic or negatively stained specimens or by atomic force microscopy. Flushed viruses were able to bind to their host cell surface and enter endosomes, suggesting that the fusion with the endosomal membrane was halted. The eluted rosin from the functionalized surface demonstrated its ability to inactivate viruses, indicating that the antiviral efficacy relied on the active leaching of the antiviral substances, which acted on the viruses coming into contact. The rosin-functionalized plastic thus serves as a promising candidate as an antiviral surface for enveloped viruses.

**IMPORTANCE**  During seasonal and viral outbreaks, the implementation of antiviral plastics can serve as a proactive strategy to limit the spread of viruses from contaminated surfaces, complementing existing hygiene practices. In this study, we show the efficacy of a rosin-functionalized plastic surface that kills the viral infectivity of human coronaviruses within 15 min of contact time, irrespective of the humidity levels. In contrast, non-functionalized plastic surfaces retain viral infectivity for an extended period of up to 48 h. The transient attachment on the surface or the leached active components do not cause major structural changes in the virus or prevent receptor binding; instead, they effectively block viral infection at the endosomal stage.

**KEYWORDS**  antiviral surface, virus persistence, human coronavirus, plastic, tall oil rosin

During the past few decades, the threat of infectious diseases has been on the rise. A recent example is the COVID-19 pandemic, which was caused by the severe acute respiratory syndrome coronavirus 2 (SARS-CoV-2) causing high mortality (1). Development of strategies to combat the transmission of these contagious pathogens is currently a global priority. During the COVID-19 pandemic, preventive strategies like the use of face masks, travel restrictions, surface disinfection, hand sanitization, and mass vaccination were used (2). Vaccines reduced morbidity and mortality to a large extent but did not help eliminate the virus. Thus, to complement these strategies, we

Address correspondence to Varpu Marjomäki, varpu.s.marjomaki@jyu.fi.

The authors declare no conflict of interest.

See the funding table on p. 22.

need novel antiviral surfaces that lower the virus load directly on contaminated surfaces, decreasing the probability of virus transmission.

Respiratory viruses, like coronaviruses, are transmitted mainly through large droplets and small aerosols from infected people when they breathe, cough, sneeze, and converse. Every cough or sneeze produces about 3,000 or 40,000 droplets, respectively, with an average viral load of $7 \times 10^6$ copies/mL, subject to the size of the droplet, severity of infection, and type of respiratory activity (3). Smaller aerosols can stay in the air for a longer duration and actively transmit the virus, while larger droplets quickly settle on inanimate objects and contaminate the surface (3). Contaminated fomite surfaces can be classified into two categories, i.e., porous and non-porous surfaces. Studies have suggested that viruses stay infectious on porous surfaces only from minutes to hours due to the capillary action of the pores and faster evaporation rate from the surface (4). On the other hand, viruses may stay active on non-porous surfaces for days to several weeks (5–7).

To reduce the viral load on non-porous surfaces like plastic, several disinfectants or surfactants, such as quaternary ammonium, sodium hypochlorite, and hydrogen peroxides, are used (8). However, due to the health hazards associated with the regular use of strong disinfectants, laborious repetitive cleaning, crazing in plastics, and a direct, negative impact on biodiversity, we need safer and more sustainable options (9, 10). Some strategies have been developed in the past to design self-cleaning plastics with anti-microbial properties (11). However, these methods can be expensive and toxic to humans and the environment (12). An alternate approach is to explore natural organic products as potential antivirals.

Coniferous tree-derived resin and rosin compounds have been explored for their therapeutic potential in the past (13). Rosins come from tall oil, which is a side stream product obtained by a process called kraft pulping of coniferous tree barks (14). Tall oil contains a mixture of 5%–50% fatty acids, 15%–55% resin acids, and 5%–35% unsaponifiable and neutral materials. Rosin components have demonstrated excellent anti-inflammatory, antitumor, antifungal, and antibacterial activities (15–19). The antiviral nature of rosin and its components has also been recognized, for example, against influenza, respiratory syncytial virus, SARS-CoV-2, and herpes simplex virus (20–22). Despite the widespread use of rosin compounds in various industrial applications, such as adhesives, rubbers, paints, coatings, soaps, detergents, paper cutting, varnishes, and emulsions, their potential as antiviral agents in applications has been relatively overlooked (14).

In our study, we show that a plastic surface functionalized with tall-oil rosin effectively inactivates the seasonal human coronavirus OC43 (HCoV-OC43) and the more virulent SARS-CoV-2. Through molecular virology assays and different imaging modalities, we demonstrate that the rosin-functionalized plastic leads to efficient loss of virus infectivity by actively releasing rosin.

## MATERIALS AND METHODS

### Cells

Human lung fibroblast (MRC-5) cells and green monkey kidney epithelial (Vero E6) cells were both obtained from the American type of culture collection (ATCC, Manassa, VA, USA). Both cell lines were propagated in Eagle's Minimum Essential Medium (MEM) (Gibco, Paisley, UK) supplemented with 10% fetal bovine serum (Gibco, Paisley, UK), 1% L-GlutaMAX (Gibco, Paisley, UK), 1% antibiotics (penicillin/streptomycin) (Gibco, Paisley, UK) and stored in a humidified 5% $CO_2$ incubator at 37°C.

### Viruses

Severe acute respiratory syndrome coronavirus 2 used in our experiments was kindly provided by the University of Helsinki. The SARS-CoV-2 virus (SARS-CoV-2/Finland/1/2020) was an isolate from the first COVID-19 patient in Finland (GenBank:

MT020781.2) (23). HCoV-OC43 (ATCC VR1558) was obtained from the American Type Culture Collection (Manassa, VA, USA). The virus was cultured in MRC-5 cells containing 2% MEM with a multiplicity of infection (MOI) of 2. The inoculum was replaced after 2 h of infection with fresh 2% MEM. The supernatant containing the cultured crude virus was collected 2 days post-infection (p.i.). The cell debris was pelleted using a swing-out rotor at 2,700 × $g$ for 3 min at room temperature (RT) (Heraeus Megafuge 1.0 R, Germany). The supernatant was flash frozen and stored as a crude working stock at −80°C.

### Purification of HCoV-OC43 virus

The purification protocol for the HCoV-OC43 virus was adapted and modified from Dent et al. (24). A subconfluent layer of MRC-5 cells cultivated in a 175 cm$^3$ flask was infected with HCoV-OC43 (ATCC) at an MOI of 3 for 72 h, after which the supernatant was collected and the cell debris was pelleted by centrifugation at 10,000 × $g$ at 4°C for 20 min. The virus in the supernatant was precipitated using 10%, wt/vol polyethylene glycol 6000 and 2.2% of NaCl (25). The solution was stirred for 30 min at 4°C, after which the precipitate was centrifuged at 10,000 × $g$ at 4°C, for 30 min. The pellet was dissolved in 3 mL HEPES saline buffer [1 mM HEPES pH 6.7, 0.9% NaCl (wt/vol)] and stored on ice. The solution containing the virus appeared viscous in consistency. The virus was concentrated by pelleting through a stepwise sucrose gradient (10%–20%–30%) at 100,000 × $g$ at 4°C for 2 h. The gradient was prepared by adding 3 mL of 30% sucrose to the bottom, followed by the same amount of 20% and then 10% on the top. The pellet was dissolved in 100 µL of cold HEPES saline buffer and stored at −80°C. The infectivity of the virus batch was calculated using the end-point titration method (26).

### Plastic samples

We used two types of plastic surfaces for our study: low-density polyethylene (standard LDPE) and PREXELENT, generously provided by Premix Ltd. The plastic surfaces were dimensioned as 1 cm$^2$. PREXELENT is a functional plastic that is prepared by incorporating 10 wt% of rosin extracted from coniferous trees into standard LDPE (patent WO2018229190A1). Throughout our article, we will refer to PREXELENT as rosin-functionalized plastic and in the figure sets as rosin. Standard LDPE was chosen as a control to compare the results of the rosin-functionalized plastic. "Conifer pitch," which is the active ingredient inside the rosin-functionalized plastic, was also provided in powder form by Premix Ltd. Additionally, a variety of industrial grade plastic varieties like acrylonitrile butadiene styrene (ABS), polyamide (PA6), polycarbonate (PC), polypropylene (PP), and polymethyl methacrylate (PMMA) were obtained via Aikolon Ltd. Prior to the experiments, all samples were sterilized with 70% ethanol for 30 s.

### Infectivity of HCoV-OC43 studied on different surfaces using a cytopathic effect inhibition assay and RT-qPCR

The persistence of HCoV-OC43 and the antiviral effect of the rosin-functionalized plastic against HCoV-OC43 were studied using the cytopathic effect (CPE) inhibition assay, modified from Schmidtke et al. (27). The experiments were conducted following the international standard ISO 21702, which is a test method to determine the antiviral activity on plastic surfaces, with slight modifications. Ethanol-sterilized plastic samples were added to a 12-well plate with relative humidity (RH) of ~92%, maintained using 6 mL of ddH$_2$O. The RH maintained inside the plate assembly was measured with a commercial RuuviTag wireless sensor (Ruuvi Ltd, Finland). MRC-5 cells were seeded at a density of 15,000 cells/well in a 96-well flat-bottomed microtiter plate (Sarstedt, Numbrecht, Germany) and allowed to grow overnight (O/N) in an incubator at 37°C, with 5% CO$_2$. On the following day, a droplet of 5 µL of HCoV-OC43 virus (1.60 × 10$^4$–10$^5$ PFU) was loaded onto the different plastic surfaces and incubated for different time points (24, 48, and 72 h) at RT and ~92% RH inside the 12-well plates. We used several controls in the experiment (i) an equivalent amount of virus without any surface treatment was used as a positive control. (ii) MRC-5 cells without any virus infection were used as a negative

control (or mock infection). (iii) Additionally, a sample control without the addition of the virus was used to test the cytotoxicity of the surfaces on MRC-5 cells. A glass coverslip was gently placed on the virus-loaded surfaces to ensure maximum contact of the virus with the surface. At the end of each incubation time point, the surface was rinsed with 995 µL of 2% MEM, followed by gentle rocking of the 12-well plate for 1 min to facilitate the detachment of the virus from the surface. The medium containing the flushed virus was added to the cultured MRC-5 cells (MOI: 0.01–0.1) in a 96-well plate. The cells were incubated for 5 days in a humidified 5% $CO_2$ incubator at 34°C until the cytopathic effect (rounding and detachment of cells) was seen. After the development of the cytopathic effect, the cells were washed twice with phosphate buffer saline (PBS), after which they were stained with a CPE stain (containing 0.03% crystal violet, 2% ethanol, and 3.5% formaldehyde in $ddH_2O$) for 12–15 min. Next, the cells were washed twice with $ddH_2O$ to get rid of excess stain, and finally, the cells were lysed for 5 min to elute the stain out of healthy cells with the help of CPE lysis buffer (0.03 M sodium citrate and 1 N HCl in 47.5% ethanol). Thereafter, the viable cells were quantified by measuring the absorbance of the stain in the 96-well plate at 570 nm (Victor X4 2030 Multilabel Reader, PerkinElmer). The amount of infectious viral RNA in the sample was also quantified using RT-qPCR from the cell culture supernatant collected 3 days p.i. The protocol for RT-PCR and qPCR has been elaborately described in "Quantification of HCoV-OC43 viral RNA directly flushed from the surfaces using RT-qPCR" section.

The effect of humidity on the antiviral function of the rosin-functionalized plastic was studied similarly as above, with the exception that the incubation of the virus-loaded surface was done inside a custom-made chamber fitted in a laminar in the BSL2 facility, capable of accurate measurements of temperature and relative humidity (Kenttäviiva Ltd, Finland). In addition, the coverslip was not used on the surface in order to allow various humidities to act on the surface and on the viruses. During the experiment, the RH was controlled to either 20% or 40% and the incubation times were shorter (5, 15, 30, and 60 min) compared to those tested earlier.

## Persistence of SARS-CoV-2 measured using RT-qPCR

The persistence studies of SARS-CoV-2 on the standard LDPE and rosin-functionalized plastic were carried out in a Biosafety level-3 unit. Vero E6 cells were grown in a 96-well flat-bottomed microtiter plate, with each well containing 50,000 cells. The cell cultures were maintained for 24 h at 37°C in an environment with 5% $CO_2$. On the following day, a 5 µL droplet ($5 \times 10^3$ PFU) of the SARS-CoV-2 was added on top of both the surfaces and incubated for different time points (5 min, 30 min, 1 h, 2 h, and 4 h) at RT and ~92% RH. At the end of each time point, the surface containing the virus was flushed with 2% MEM and gently mixed for 1 min to detach any virus from the surface. The flushed virus was then added to the Vero E6 cells at an MOI of 0.005. The cells were then placed in a humidified incubator with 5% $CO_2$ at a temperature of 34°C for a 3-day incubation period. After 3 days post-infection, the supernatant was collected and the viral RNA was extracted.

The RNA from the supernatant was extracted using the Chemagic viral DNA/RNA kit (Perkin Elmer) and the Chemagic 360 instrument. For the RNA extraction, a master mix containing Poly (A) RNA, Proteinase K, and Lysis buffer 1 was mixed in the proportion mentioned inside the kit by the manufacturer. The samples were initially diluted 10 times in $ddH_2O$. The diluted samples and the master mix were added into deep-well Chemagic plates. In a different low-well plate, 15 µL of magnetic beads was pipetted, and in a deep-well Chemagic plate, 80 µL of elution buffer was added. The RNA was extracted using the Chemagic 360 machine as per the manufacturer's instructions. The extracted RNA was stored at −80°C. After the RNA isolation, a TaqMan-based real time RT-PCR assay was used to transcribe the viral RNA, amplify it, and detect the SARS-CoV-2 genes using the SARS-CoV-2 RT-qPCR reagent kit (Perkin Elmer, Turku, Finland).

In order to compare the relative amounts of viral RNA (number of infective viruses) present in the culture supernatant after 3 days of cultivation in cells, we used the Cq

values obtained from the qPCR experiment. Cq values obtained from the qPCR reaction of the virus flushed from the standard LDPE and rosin-functionalized plastic surfaces were placed into the following self-derived equation to obtain the difference in the RNA amount, ΔRNA:

$$\Delta RNA = 0.9646e^{0.6948 \cdot \Delta Cq}$$

where ΔCq is the Cq value difference between the mean of the rosin-functionalized plastic surface to the mean of the standard LDPE surface. A log value of ΔCq was then calculated to describe the difference in virus amounts.

## Statistical analysis

The results from the CPE assays were plotted as a column graph of cell viability, and the statistical analysis was performed using GraphPad Prism software 6 (GraphPad Software, San Diego, CA, USA). The CPE results were normalized against the mock infection and are presented as mean + standard error of the mean (SEM). The statistical significance was calculated by performing a one-way analysis of variance (ANOVA) of all test samples against the virus control followed by the Bonferroni tests (*$P < 0.05$, **$P < 0.01$, ***$P < 0.001$, and ****$P < 0.0001$).

## Quantification of HCoV-OC43 viral RNA directly flushed from the surfaces using RT-qPCR

A droplet of 5 µL of HCoV-OC43 ($1.60 \times 10^5$ PFU) was loaded onto the standard LDPE and rosin-functionalized surface and incubated for different time points (1, 5, 15, and 30 min and 1, 2, and 4 h) at RT, ~92% RH inside 12-well plates. A similar amount of virus without any surface treatment was used as a positive control. At the end of each time point, the virus on the surface was rinsed with 995 µL of culture media (2% MEM) and gently agitated for 1 min to facilitate the detachment of the virus from the surface. RT-qPCR was then used to measure the amount of viral RNA in the flushed samples. Processing of the flushed samples to measure the viral RNA was done through the following steps: (i) the samples were diluted five times with RNase-free water (J71768, Thermo Fisher Scientific, Geel, Belgium) and heat treated for 5 min at 75°C to release the RNA from the viruses. (ii) Reverse transcription (RT') was carried out to make cDNA from the viral RNA using 20U M-MLV reverse transcriptase enzyme (Promega, WI, USA), RT-buffer (Promega, WI, USA), RNase-free water (J71768, Thermo Fisher Scientific), 4U RNAsin ribonuclease inhibitor (Promega, WI, USA), 0.5 mM dNTPs (Promega, WI, USA), and 1.2 µM reverse primer (5′-AATGTAAAGATGRCCGCGTATT) (Merck, Darmstadt, Germany). The RT' reaction was carried out for 1 h at 42°C, and then the RT' enzyme was inactivated at 70°C for 10 min. (iii) Finally, the cDNA was amplified using qPCR. A master mix containing SYBR Green Supermix (BioRad), 600 nM forward (5′-TGTTAGGCCRATAATTGAGGAC) (Merck, Darmstadt, Germany) and reverse primers (5′-AATGTAAAGATGRCCGCGTATT) (Merck, Darmstadt, Germany) and the RNase-free water was prepared as per the manufacturer's protocol. cDNA template (5 µL) from the RT' reaction was added to the prepared master mix and run through the Touch Thermal Cycler (BioRad C100, BioRad, Helsinki, Finland). The amplification steps were as follows: 10 min at 95°C, 40 cycles of 15 s at 95°C and 1 min at 50°C, 5 s at 72°C, 1 min at 95°C, followed by cooling at 12°C for 10 min.

## Transmission electron microscopy

Purified HCoV-OC43 (10 µL of $1.09 \times 10^8$ PFU) was added on the sterilized standard LDPE and rosin-functionalized plastic and incubated for 1 h at RT with ~92% RH. Post-incubation, 10 µL of PBS was added to the surface and the droplet was gently mixed. A virus control of the same amount without any surface treatment was also prepared. Formvar-coated copper grids were glow discharged (EMS/SC7620 mini-sputter coater), and negative staining was done as previously described (28). Briefly, 5 µL of virus (control,

standard LDPE, and rosin-functionalized plastic) was added onto the copper grids for 30 s and blotted with Whatman paper (Whatman 3 MM). The samples were negatively stained using 1% (wt/vol) phosphotungstic acid for 10 s and blotted. The samples were dried O/N before imaging in a JEOL JEM-1400 transmission electron microscope (JEOL, Tokyo, Japan) equipped with a $LaB_6$ filament. The images were captured using a bottom-mounted Quemesa $4,008 \times 2,664$-pixel CCD camera.

## Atomic force microscopy

A volume of 10 µL of purified HCoV-OC43 virus ($1.09 \times 10^7$ PFU) was deposited on the standard LDPE and rosin-functionalized plastic surfaces and incubated for 1 h at RT with a RH of ~92%. Post-incubation, 10 µL of 2.5% glutaraldehyde (GA) was added to fix the virus and maintain the morphology. Post-fixation, the droplet containing the virus was transferred onto the silicon surface that was pretreated with air (oxygen/nitrogen) plasma (Electron Microscopy Sciences, Hatfield, PA, USA) and 0.01% poly-l-lysine for 15 min to enhance the attachment of the viruses to the surface. Virus without any surface treatment was used as a control and was added directly onto poly-l-lysine-coated silicon chips for atomic force microscopy (AFM) measurements. Bruker Dimension Icon atomic force microscope (Bruker Corporation, Billerica, MA, USA) and Scanasyst fluid tips (nominal radius 20 nm) were used during the liquid AFM measurements. Peak force quantitative nanomechanical mapping mode was used in all the measurements. Imaging was done in PBS (scan size of 5 µm, scan rate varying between 0.332 and 0.939 Hz, and peak force amplitude between 100 and 200 nm). The peak force setpoint was kept below 1 nN and the scan rate, peak force, and peak force amplitude were adjusted to minimize the forces between the tip and the virus to obtain the best image quality. Bruker analysis software 1.9 version was used to plot and analyze the data. In addition, the data were plotted and tested for statistical significance in Origin Pro 2022. Viruses that appeared in large clusters were excluded from the measurements.

## Helium ion microscopy

A 5 µL droplet of HCoV-OC43 virus ($5.45 \times 10^6$ PFU) was loaded onto the standard LDPE and rosin-functionalized plastic surfaces on ice and fixed in 2 mL of 2.5% GA + 0.1 M sodium cacodylate (NaCaC) solution for 30 min. After three washes with 0.1 M NaCaC solution, 3 min each, the surfaces were stained for 30 min with 1% $OsO_4$ prepared in 0.1 M NaCaC. After two washes with 0.1 M NaCaC solution, the surfaces were subjected to increasing ethanol series (50%, 70%, 96%, and 99%) and dried using a critical point dryer (Leica Microsystems EM CPD300). The samples were stored in a vacuum chamber until they were taken for helium ion microscopy (HIM) (Zeiss Orion Nanofab) imaging, where the acceleration voltage of 30 kV and an aperture of 10 mm were used. Spot size 6 was used, giving an ion current of 0.2 pA. It was possible to image the plastic samples without a flood gun, but on the rosin-functionalized plastic, the charge compensation was used with line averaging. Imaging was done from an 85° tilted angle. The aspect ratio of the particles was calculated by measuring their maximum height and the width of the single particles from the images.

## Hydrophobicity measurements

A water droplet of 40 µL was added on top of the standard LDPE and rosin-functionalized plastic. Images of the droplet were taken from the side and top angle using a smartphone camera (Xiaomi MI 8). The contact angle was measured from the base of the surface using Inkscape drawing software (Inkscape Project 2020).

## Cryo-electron microscopy

Purified HCoV-OC43 virus (5 µL, $5.45 \times 10^7$ PFU) was added to the surfaces and incubated for 1 h at RT with ~92% RH. Post-incubation, 5 µL of HEPES saline was added to the surface and the droplet was gently mixed. A virus control without any surface treatment

and with the same amount was also prepared. A total of 10 µL of the virus collected from the surface was exposed to ultraviolet (UV)-C irradiation at 100 mJ cm$^{-2}$ using a UVP crosslinker CL-1000 (Analytik Jena, Jena, Germany) at A$_{254}$ nm for inactivation before plunge freezing (29). The virus samples were loaded on a glow-discharged Quantifoil electron microscopy grid with 2 nm carbon film R1.2/1.3, mesh 300 and incubated for 1 min before plunging at 22°C, 85% humidity, and 1.5 s blotting time with the Leica EM GP plunger. Cryo-electron microscopy (CryoEM) data were collected using a FEI Talos Arctica field emission transmission electron microscope (TEM), equipped with a Falcon III direct electron detector at the Instruct-FI Centre's CryoEM Core Facility. Data were collected in counting mode at 150,000× nominal magnification at a 0.97 Å/pixel sampling rate using a total dose of 30 e$^{-}$ Å$^{-2}$. Movies were motion corrected using Relion's motion correction (30–36) implemented in Scipion 3.0.12 (37, 38) and the contrast transfer function was estimated using ctffind4 (39–41). Images were further adjusted in Corel Draw 2023 to improve the contrast of viral features.

## Sedimentation assay

Genome release from the virus capsid was studied using ultracentrifugation. For this experiment, 5 µL of the purified HCoV-OC43 virus ($2 \times 10^6$ PFU) was added to the surface with a glass coverslip and incubated for 1 h at RT with ~92% RH. Post-incubation, the samples were flushed with 2% MEM and collected into polyallomer centrifuge tubes (Beckman, Palo Alto, CA, USA). The flushed content was ultracentrifuged (A-95 rotor, 20 psig, 30 min) using the Beckman Coulter Airfuge centrifuge (Brea, CA, USA). A virus of the same amount, without any pretreatment on the surface, was used as an experimental control. Post-centrifugation, the supernatant was collected in a separate tube, and the pellet was redissolved in 2% MEM. The RNA from both the pellet and supernatant was isolated using the QIAamp Viral RNA Mini kit (Qiagen, Hilden, Germany, ref. 52906). The isolated RNA was converted into cDNA and amplified using RT-PCR and qPCR as described above. The cDNA was diluted 100 times before performing the qPCR.

## Binding assay

MRC-5 cells were cultured in a 6-well plate with a flat glass bottom (Fisher Scientific) at a density of 350,000 cells/well and incubated O/N at 37°C. On the subsequent day, 10 µL of the purified HCoV-OC43 virus ($1.89 \times 10^7$ PFU) was added to the surface with a glass coverslip and incubated for 1 h at RT with ~92% RH. Post-incubation, the samples were flushed with 2% MEM and the flushed virus was added to the MRC-5 cells on ice for 1 h while rocking. After three washes with 0.5% BSA/PBS, 5 min each, the 6-well plate was subjected to three cycles of freeze-thaw by alternating between −80°C and RT to detach the cells from the bottom of the plate. The viral RNA from the virus bound to cells was isolated using the QIAamp Viral RNA Mini kit (Qiagen, Hilden, Germany, ref. 52906). The isolated RNA was converted into cDNA and amplified using RT-PCR and qPCR as described above. The cDNA was diluted 1,000 times before performing the qPCR.

## Immunofluorescence and confocal microscopy

Confocal microscopy was applied to study the different stages of the virus infection cycle. Purified HCoV-OC43 virus (5 µL, $2 \times 10^6$ PFU) was added to the surfaces and incubated for 1 h at RT with ~92% RH. Viruses flushed from the surfaces were added onto cells (MOI: 100) into 96-well plates and kept at RT for 1 h, followed by 1 h at 34°C. Infection was allowed to proceed for 1 or 15 h at 34°C and then the cells were fixed with 4% paraformaldehyde in PBS. Cells were permeabilized with 0.2% Triton X-100 for 5 min, followed by antibody labeling with affinity-purified rabbit monoclonal antiserum against the S protein (1:16 diluted), mouse monoclonal antisera against human beta-tubulin (#sc-58886, Santa Cruz Biotechnology) or J2 (Scicons, Hungary) against the dsRNA. The secondary antibodies used were Alexa Fluor 555 goat polyclonal IgG against rabbit (catalog no. A-21429; Thermo Fisher Scientific) and Alexa Fluor 488 goat polyclonal IgG

against mouse (catalog no. A-21121; Thermo Fisher Scientific). The second PBS wash was supplemented with 4′,6-diamidino-2-phenylindole (DAPI) (Invitrogen/Molecular Probes, ref. D3571) (1:40,000) in PBS to stain the cellular nucleus. All antibody dilutions were made in 3% BSA in PBS. The cells were imaged using a Nikon A1R confocal microscope. Images were taken with a 40× objective and numerical aperture of 1.25. Three lasers were used: a 405 nm diode laser, a 488 nm multiline argon laser, and a 561 nm sapphire laser. The virus spike protein and dsRNA channels were adjusted with the cell control, such that the cell control had a very low signal. Several images were taken using the automation option in the software to accumulate at least 700 cells per case. The montages from the images were made with Fiji2 (ImageJ). Cell profiler 4.2.1 was used to quantify the amount of infection in the wells. The nuclei were selected as primary objects, and the infected cells were then identified as secondary objects if they were present around the previously identified primary object (nuclei). The primary objects were identified using the Otsu thresholding method. Once the primary and secondary objects were classified, the software automatically calculated the area and the intensity of the secondary objects, which were then exported as an Excel data sheet. At this point, manual thresholding was done to differentiate infected cells from non-infected cells having some background signal. The data were subsequently plotted as a column graph and represented as infection (%).

## UV-vis spectroscopy

To measure the rosin content secreted to the flush medium, rosin-functionalized plastic was flushed with ddH$_2$O for 1 h and O/N using gentle rocking. As a reference of the patented compound embedded inside the rosin-functionalized plastic, Premix Ltd. provided us with a powdered sample of elemental conifer pitch (secreted and cured oleoresin mixture) that was dissolved in 94%, vol/vol of ethanol and 10-fold serial dilutions were made to obtain a final concentration of 0.0001%, wt/vol of conifer pitch. The UV-vis spectra of the flushed material without any further dilution were measured with LAMBDA 850 UV-Vis spectrometer (Perkin Elmer Life Sciences) and Quartz cuvettes (Hellma) with optical pathlength of 1 cm. The measured raw spectra were background corrected with the measured spectrum of EtOH using Origin 2022 software.

## RESULTS

### Virus persistence and antiviral studies

#### *Rapid loss of coronavirus infectivity on rosin-functionalized plastic compared to other plastic surfaces*

The persistence of HCoV-OC43 on different plastic surfaces was evaluated at multiple time points using a CPE inhibition assay. The results showed a similar trend of virus persistence on all the plastic surfaces for the first 24 h (Fig. 1A). Highly infectious viruses were recovered from all the surfaces at 24 h. At 48 h, the amount of virus infectivity recovered from the ABS, PA6, and PC surfaces was not statistically different from the virus control; however, there were statistical differences in the virus infectivity recovered from the PMMA, PP, and LDPE surfaces. This suggests that there was some loss of virus infectivity on these surfaces. Almost full loss of infectivity of HCoV-OC43 from all the above-mentioned surfaces was seen at 72 h. To ensure that the results from the persistence of the virus come solely from the virus infectivity and not from any toxicity associated with the different plastic surfaces, we measured their cytotoxicity as well. Cytotoxicity was measured by flushing the surface with media and adding it to the cells. The results showed that none of the plastic surfaces were cytotoxic (data not shown). These findings revealed the actual extent of HCoV-OC43 persistence on various plastic surfaces.

As mentioned in the Materials and Methods section, rosin-functionalized plastic was prepared by incorporating tall-oil rosin into standard LDPE plastic through a patented

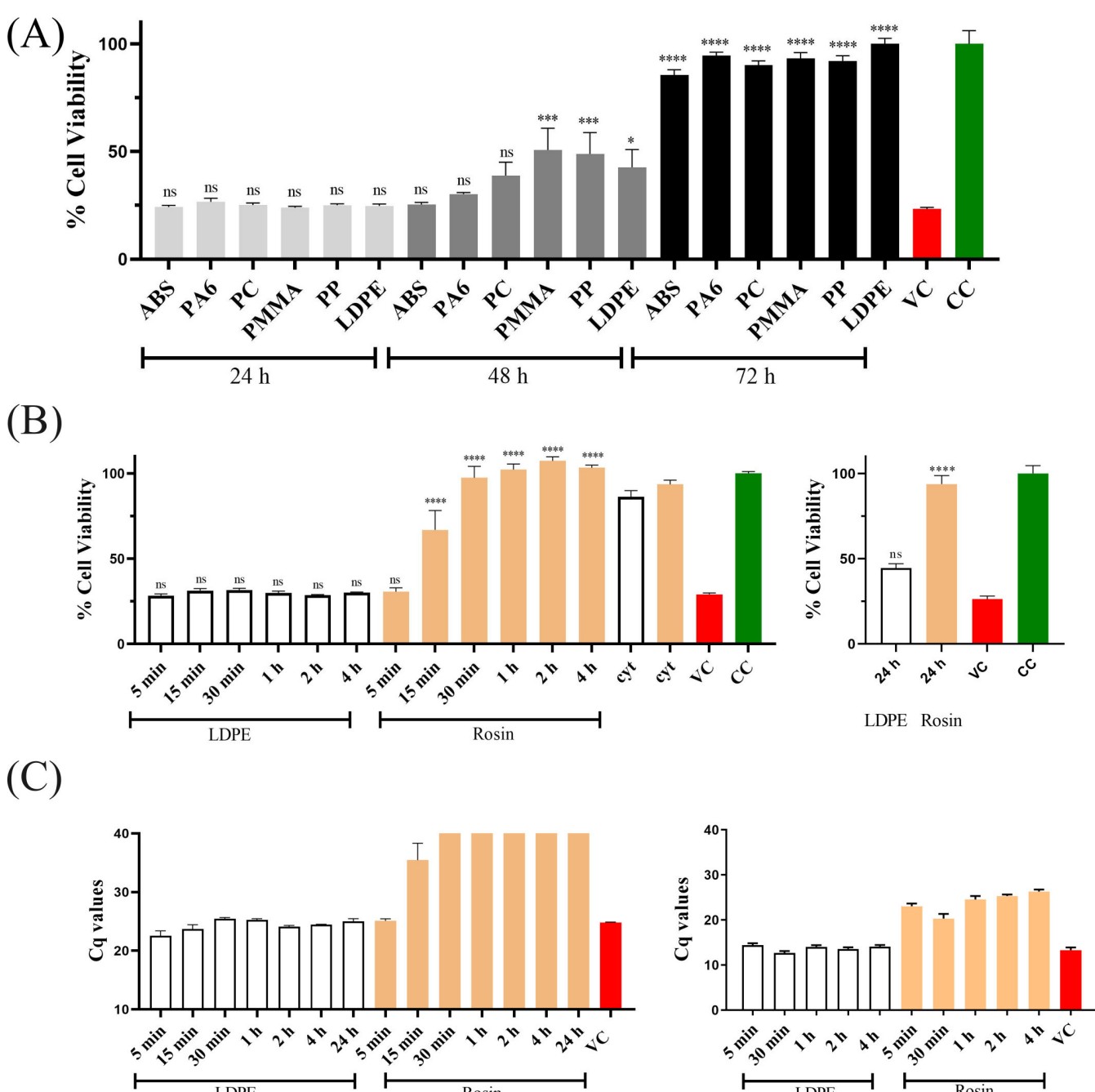

FIG 1 Persistence of HCoV-OC43 on (A) different plastic surfaces measured up to 72 h. Virus titer on various plastic surfaces—$1.6 \times 10^4$ PFU. (B) Standard LDPE and rosin-functionalized plastic measured up to 24 h of contact time with the surface. Virus titer on standard LDPE and rosin-functionalized plastic surface—$1.6 \times 10^5$ PFU. In both (A) and (B), a CPE assay was conducted to assess cell viability post-infection, and the treated samples and virus control were normalized against the cell control. The experiments were replicated three times, with each condition having three technical replicates. (C) Infectivity of HCoV-OC43 and SARS-CoV-2 on standard LDPE and rosin-functionalized plastic, measured up to 24 and 4 h of contact time, respectively, using RT-qPCR (see Materials and Methods). The qPCR experiment was conducted twice, each with three technical replicates. For all panels (A through C), the results are presented as average values + SEM. Statistical analysis employed one-way ANOVA, followed by the Bonferroni test (*$P < 0.05$, ***$P < 0.001$, and ****$P < 0.0001$). ns, not significant; VC, virus control; CC, cell control; and cyt, cytotoxicity.

technology (Patent no. WO2018229190A1). Industrial grade standard LDPE was selected as one of the controls in all our experiments to enable comparison of the results with those obtained using the rosin-functionalized plastic. The persistence of the HCoV-OC43

virus on the rosin-functionalized plastic was also tested at different time points. From the CPE assay result (Fig. 1B), there was a gradual loss of virus infectivity starting at 15 min and a complete loss of infectivity at 30 mins. Even with an extended incubation of 24 h on the surface, the antiviral effect of the rosin-functionalized plastic surface against HCoV-OC43 was sustained. In contrast, the virus on the standard LDPE surface remained infectious until the maximum time point tested (24 h). The cytotoxicity results of the standard LDPE and rosin-functionalized plastic indicated that they were both non-toxic toward MRC-5 cells (Fig. 1B).

To compare the relative amount of infectious viral RNA recovered from the standard LDPE and rosin-functionalized plastic surfaces for HCoV-OC43 and SARS-CoV-2, we collected the supernatant from virus-infected cells after 3 days p.i. and performed qPCR analysis, as shown in Fig. 1C. The results were presented as bar graphs of the quantification cycle (Cq) values. Cq values are inversely related to the amount of RNA in the sample, meaning higher Cq values correspond to lower detected viral RNA levels. For both HCoV-OC43 and SARS-CoV-2, the Cq values of the virus samples flushed from the standard LDPE surface were quite similar to the virus control. This indicated that highly infectious virus was recovered from the standard LDPE surface at all time points. In the case of HCoV-OC43, low Cq values were reported for the virus sample recovered from the rosin-functionalized plastic surface after 5 min of incubation. Beyond this initial period, Cq values from virus samples were much higher after 15 min and undetectable after 30 min and later, indicating a rapid decline in viral titers on the rosin-functionalized plastic surface. A similar pattern was observed for SARS-CoV-2, except the virus was inactivated even earlier, within 5 min on this surface.

Using the Cq values, we calculated the logarithmic reduction in virus infectivity on the rosin-functionalized plastic surface compared to standard LDPE. For HCoV-OC43, there was a 3.54 logarithmic reduction of infective viruses after just 15 min of contact with the rosin-functionalized plastic surface. Since Cq values were undetectable for subsequent time points, we considered the upper limit of detection for the qPCR machine (Cq value of 40) and estimated a reduction of more than 4 logs. For SARS-CoV-2, there was a 2.59 logarithmic reduction within 5 min of contact with the rosin-functionalized surface. Thereafter (up to 4 h), infectivity continued to decrease, reaching a reduction of 3.67 logs.

### Humidity does not affect the antiviral efficacy of rosin-functionalized plastic

The previous assays to determine the antiviral activity of the rosin-functionalized plastic surface were performed at exceedingly high RH (over 90%), as per the ISO standards. However, we were also interested to know if lower humidity would have any effect on the antiviral activity of the surfaces containing rosin compounds. For this, the virus persistence at lower RHs (20% and 40%) was studied. The results depict no major difference in the infectivity of the viruses recovered from the standard LDPE surface compared to the virus control at 20% or 40% RH (Fig. 2A and B). However, there was a difference in the persistence of HCoV-OC43 on the rosin-functionalized plastic at lower humidity (20% and 40%) compared to higher humidity (over 90%). At the lower humidities, the virus lost its infectivity already at 15 min of contact time (Fig. 2A and B) with the rosin-functionalized plastic, whereas at higher humidity, the virus was completely inactivated only after 15 min (Fig. 1B). Altogether, these results indicate that at higher humidity, the virus can persist for a longer period, delaying its inactivation on the rosin-functionalized plastic.

### Antiviral mechanism of action and virus morphology studies

#### Viruses do not bind strongly on the rosin-functionalized plastic surface

One of the plausible explanations for the loss of virus infectivity could be the strong interaction/binding between the rosin-functionalized plastic and the virus. To study this, an experiment was designed to recover the virus from the surface by gentle flushing and then quantifying the relative amounts of viral RNA in the flushed media using qPCR. The

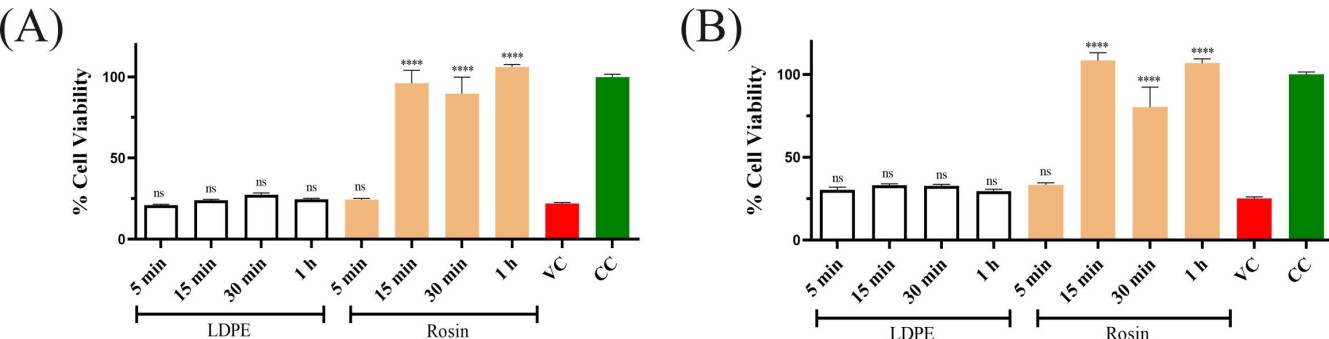

**FIG 2** The impact of lower humidity (A) 20% and (B) 40% on the antiviral activity of the rosin-functionalized plastic surface against HCoV-OC43. The experiments were conducted three times, with each condition including three technical replicates. The CPE results are presented as average values + standard error of the mean. Statistical analysis utilized one-way ANOVA, followed by the Bonferroni test (****$P < 0.0001$). ns, not significant; VC, virus control; and CC, cell control.

results showed that the media flushed from the standard LDPE and rosin-functionalized plastic contained a similar amount of viral RNA. This amount was also comparable to the virus control, which was not added on any surface (Fig. 3). This result indicated that the virus did not strongly adhere to either of the surfaces and the reason for the loss of infectivity on the rosin-functionalized plastic was not surface binding.

### AFM and TEM studies show no apparent changes in the coronavirus structure

Another reason for the loss of virus infectivity could be that the rosin from the functionalized plastic might have caused some structural changes to the virus. TEM and AFM were used to study these changes. Negatively stained images of the viruses collected from the standard LDPE and the rosin-functionalized plastic appeared similar to that of the control viruses (Fig. 4A). Viruses appeared mostly spherical to elliptical in shape with finger-like projections (spikes) originating from its envelope. In addition, the center of the virus appeared darkly stained. The mean diameter of the viruses was calculated to be an average of 110 nm.

For AFM studies, the flushed viruses were transferred onto a silicon surface to ease imaging while using the tapping mode in liquid. Viruses flushed from the surfaces were found to bind very quickly to the silicon surface because of poly-l-lysine, which makes the surface hydrophilic. AFM studies showed that the virus was spherical to

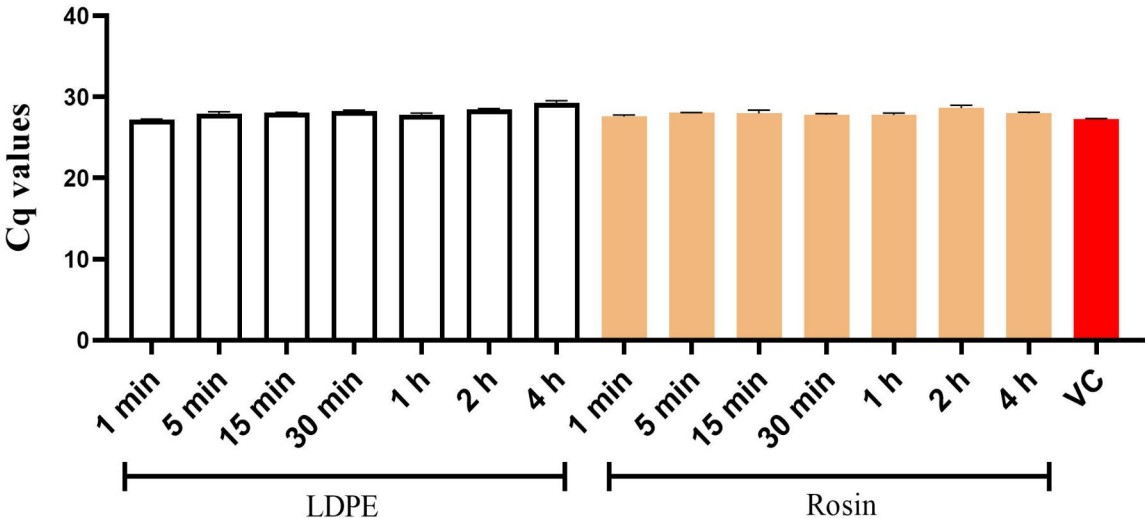

**FIG 3** Detection of HCoV-OC43 viral RNA using qPCR from samples flushed from standard LDPE and rosin-functionalized plastic, measured at various contact time points. The experiments were conducted twice, including three technical replicates. VC, virus control.

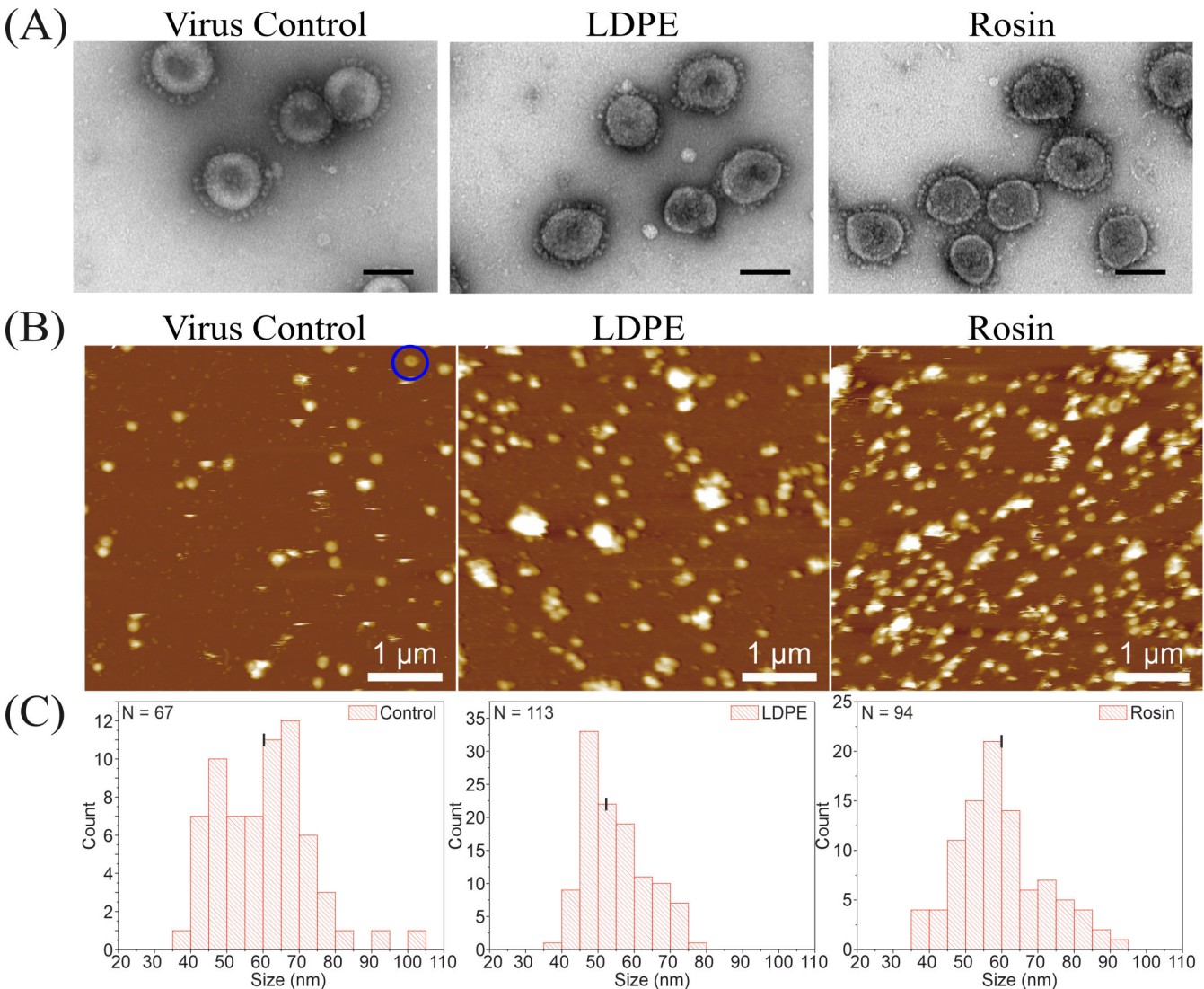

**FIG 4** Studying the impact of rosin-functionalized plastic and standard LDPE on the structure of HCoV-OC43 using (A) TEM and (B) AFM in liquid. The scale bar corresponds to 100 nm and 1 µm in the TEM and AFM images, respectively. In panel B, the blue circle highlights a doughnut-shaped virus. (C) The histogram derived from the AFM images illustrates the average size distribution of the height of individual viruses after being flushed from their respective surfaces. The average mean heights for the control virus, viruses flushed from standard LDPE, and rosin-functionalized plastic were 60.1 ± 1.6, 54.7 ± 0.8, and 59.9 ± 1.3 nm, respectively.

ellipsoidal, which was consistent with the TEM results. Also, most of these viruses seemed to be present as single particles (Fig. 4B). Sometimes a doughnut-shaped virus was also observed, and an example of one is highlighted in a blue circle in the control virus image (Fig. 4B). A histogram (Fig. 4C) depicting the height distribution of single viruses flushed from the standard LDPE, rosin-functionalized plastic, and control virus was plotted. The black lines (in the histogram) depict the mean value of the height of the virus, which were calculated as 60.1 ± 1.6, 54.7 ± 0.8, and 59.9 ± 1.3 nm for the control, standard LDPE, and rosin-functionalized plastic samples, respectively. The size distribution of the viruses was wider in the case of control and rosin-functionalized plastic compared to the standard LDPE sample, but the mean sizes remained similar for all samples. Overall, no major changes in the virus structure were observed after being recovered from either the standard LDPE or rosin-functionalized plastic.

### Coronaviruses appear flatter on rosin-functionalized plastic compared to that on standard LDPE

Until this stage, structural studies focused on the viruses that were flushed from the surfaces. However, it was equally important to observe the virus in its native form while it was still present on those surfaces. To accomplish this, HIM was employed, which allows for the visualization of material topology without the need for additional metal coating. As seen in Fig. 5A (i) and (ii), the standard LDPE and rosin-functionalized plastic are densely populated with about 70 nm sized virus particles. The density of the virus particles between the surfaces was quite similar, but, on the rosin-functionalized plastic, the viruses appeared to be flatter compared to those on the standard LDPE surface. To quantify this, width-to-height aspect ratios were measured from the HIM images (data not shown). Ratio of the virus on the rosin-functionalized plastic result was 2.5 ± 0.5 (S.D.) and on the standard LDPE surface was 1.6 ± 0.3(S.D.). These results demonstrated that adding the virus on top of the rosin-functionalized plastic resulted in significantly flatter virus particles. Images of the plain LDPE and rosin-functionalized plastic surfaces were taken as controls to show that the bumps seen on the surfaces were contours of the viruses and not of any surface artifacts/irregularities (data not shown).

It was important to understand whether the flattening out of the viruses on the functionalized surface was due to the hydrophilic/hydrophobic nature of the surface. For this, we took images of a droplet of water on the standard LDPE and rosin-functionalized plastic and measured the contact angle of the droplet on the surface using the Inkscape software (Fig. 5B). The contact angle measured for the standard LDPE surface was 95° and for rosin-functionalized plastic was 55°. These results suggested that the rosin-functionalized plastic is more hydrophilic than the standard LDPE surface and gives support to the reasoning for flattening of the viruses on the functionalized surface.

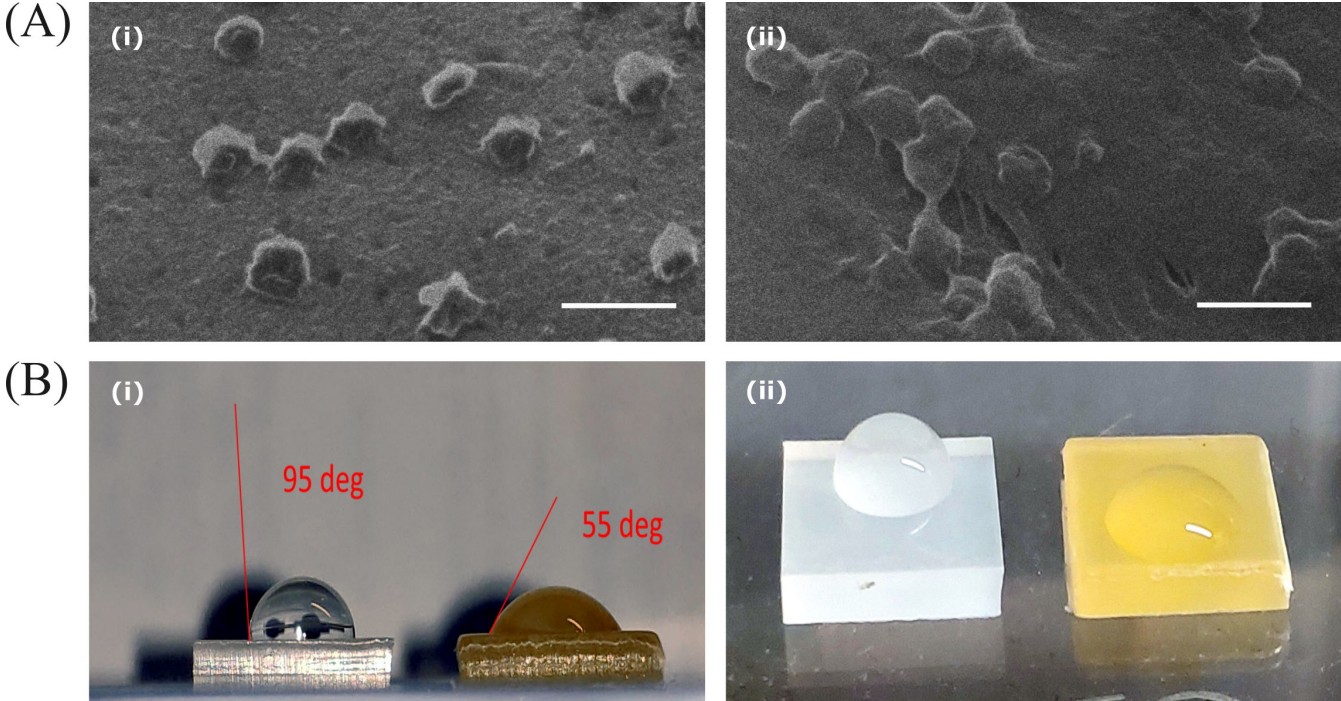

**FIG 5**  (A) HIM images of HCoV-OC43 on (i) standard LDPE and (ii) rosin-functionalized plastic. The viruses underwent a 1-h treatment on these surfaces before imaging with the HIM. Scale bar: 200 nm. (B) Camera-captured images of a water droplet on the surfaces of standard LDPE (left) and rosin-functionalized plastic (right) to measure the droplet contact angle. (i) Side view and (ii) top view.

## HCoV-OC43 flushed from the rosin-functionalized plastic retains its spikes and genome

High-resolution cryoEM was also used to examine the ultrastructure of HCoV-OC43 post-treatment on the standard LDPE and rosin-functionalized plastic surfaces. The virus flushed from both these surfaces looked structurally similar to the virus control (Fig. 6A). It retained its spikes on the surface, primarily in the prefusion state, and the nucleocapsid inside the virion was clearly visible containing the genome inside the viral envelope. Sedimentation assay was also performed to detect the presence of the genome using ultracentrifugation. The viral RNA in the pellet was quantified using qPCR. The qPCR results (Fig. 6B) showed that the amount of viral RNA recovered from the viruses flushed from the standard LDPE (Cq 21.04) and rosin-functionalized plastic (Cq 22.75) was comparable to the virus control (Cq 20.96), suggesting that the genome of the virus flushed from these surfaces was present inside the envelope. The sedimentation assay results confirmed the presence of genome inside virions, a representative of which can be seen by cryoEM.

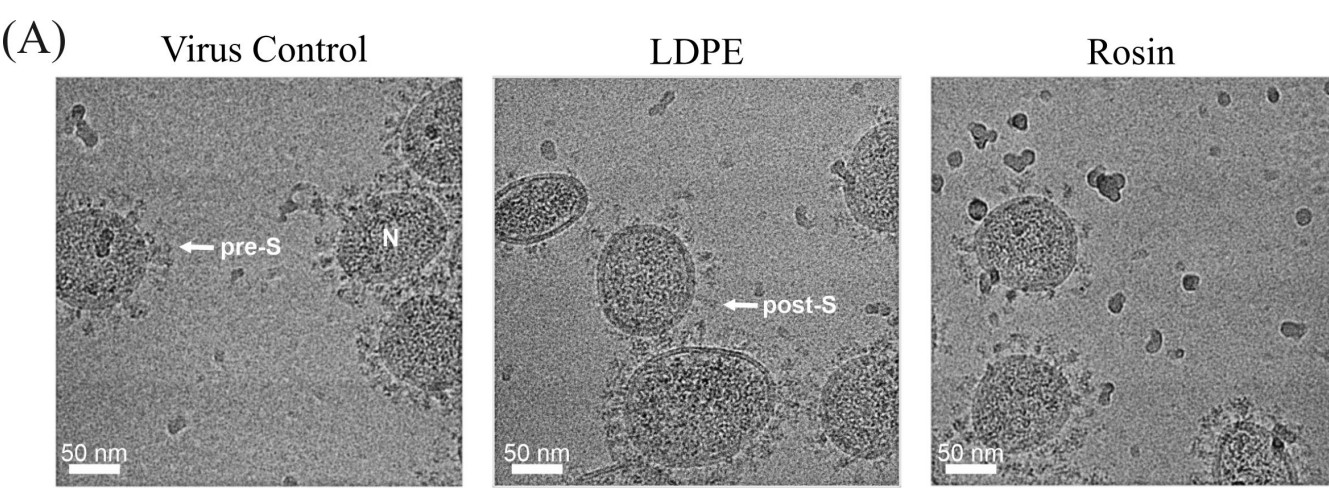

N - nucleoprotein with genome
pre - S - pre fusion S protein
post-S - post fusion S protein

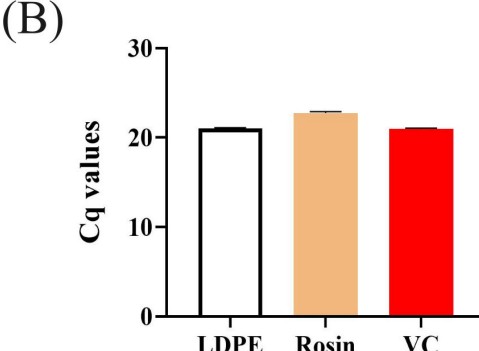

FIG 6  Investigation of HCoV-OC43 viral genome release through (A) CryoEM and (B) qPCR. (A) Representative CryoEM micrographs of virus control and from viruses flushed from LPDE and rosin-functionalized plastic. The arrows indicate a prefusion conformation spike, while "N" indicates the nucleocapsid containing the genome. Over 700 images (each with two to three viruses) were examined and analyzed to study the viruses flushed from the rosin-functionalized plastic surface. (B) Viruses flushed from both the surfaces and control viruses were all sedimented by airfuge, and the amount of pelleted RNA was measured using qPCR.

### The active component inside the functionalized surface leaches out of the surface and contributes toward the antiviral activity

Absorption spectroscopy was performed to the flushed material in comparison to the reference compound (kindly provided by Premix Ltd) to study if the material leaches significant amounts of rosin from the surface. We measured the absorption spectra of three samples, 1 h flush from the rosin-functionalized plastic, O/N flush from the rosin-functionalized plastic, and the reference compound (0.0001, wt/vol conifer pitch). The 1 h and O/N-flushed samples had RNase-free water as a solvent, and the reference compound was dissolved in 94%, vol/vol ethanol. The data in the graph in Fig. 7A have been plotted after normalizing the O/N-flushed sample with the 1 h sample. The O/N and 1-h-flushed samples show remarkably similar spectra, but they deviated slightly from the absorption spectrum of the reference sample. The reference sample shows hardly any absorption near 300 nm unlike the flushed samples with a broad absorption band at about 310 nm. The flushed samples have two absorption bands at 245 and 220 nm, like the reference sample with maxima near 240 and 216 nm. The small difference to the reference sample is likely due to some additives/fillers that the company uses to embed the rosin inside the plastic surface. Another explanation is that the molecules behave differently in different solvents due to the difference in solubility, and thus, they absorb differently due to different extinction coefficients.

To confirm that the active ingredient from the functionalized surface is responsible for the antiviral activity, we evaluated the 1 h flushed media against the HCoV-OC43 virus. From the CPE results (Fig. 7B), we could see that the flush was able to effectively rescue the cells from the virus infection. Both these results suggest that the antiviral activity of the functionalized surface comes from the leaching of the active ingredient to the surface and a direct interaction with the virus.

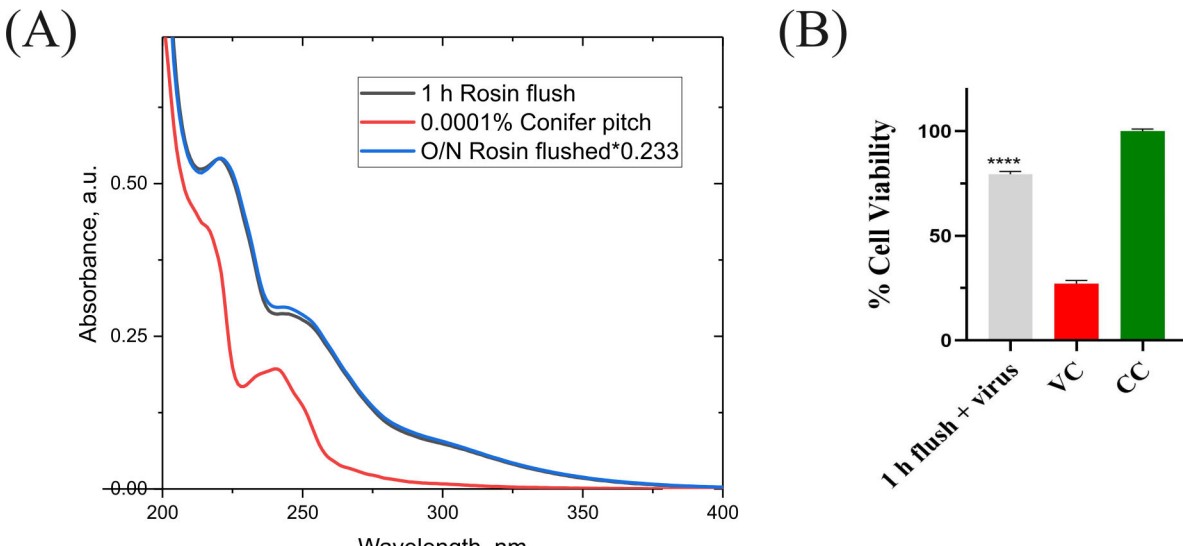

**FIG 7** The active component inside the functionalized surface leaches out into the media and can be detected using UV-vis spectroscopy and can inhibit the HCoV-OC43 virus infection. (A) The spectrum of the O/N rosin-functionalized plastic flushed sample (blue line) was normalized with the 1 h rosin-functionalized plastic flushed sample (black line). The reference spectrum of 0.0001% wt/vol conifer pitch in EtOH sample is shown in red. (B) Evaluation of the antiviral activity of 1 h flushed media against HCoV-OC43. The experiment was done once with three technical repeats. Results are plotted as average values + SEM. Statistical analysis employed one-way ANOVA, followed by the Bonferroni test (****$P < 0.0001$). VC, virus control and CC, cell control.

## Effects on the HCoV-OC43 infection cycle

### *Viruses flushed from the rosin-functionalized plastic can enter cells, but the infection does not proceed beyond the endosomal stage*

As there were only minor changes in the virus structure despite a complete loss of infection, we wanted to study if the virions were able to still bind to their receptors and enter cells. To study this, an *in vitro* binding assay was performed. The results from the qPCR-based binding assay (Fig. 8A) demonstrated that the viruses flushed from the standard LDPE and rosin-functionalized plastic surfaces were able to bind to its host cells similarly to the control virus, suggesting that the spikes in the coronavirus surface were intact to promote binding to its host cell receptors. We further confirmed the virus binding and internalization using confocal microscopy. The viruses flushed from the standard LDPE and rosin-functionalized plastic surfaces were added onto MRC-5 cells and allowed to bind for a certain amount of time before they were fixed and immuno-labeled for imaging. Normal binding to the host cells was followed by internalization over a period of 2 h (Fig. 8B). The internalized virus appeared to be lining the cellular boundaries as well as accumulating in small vesicles close to the plasma membrane. There was no apparent difference observed between the viruses flushed from the surface samples and the virus control. These results support the previous TEM, cryoEM, and sedimentation assay results that the viruses were intact, could promote receptor binding through the presence of prefusion state spikes on the viral surface, and enter cells (Fig. 4A, 6A, and B).

To study the fate of the bound virus, in another experiment, the virus was allowed to internalize for 15 h inside the cells after 1 h of binding at RT. Post 15 h, the cells were similarly fixed, permeabilized, and labeled. From Fig. 9A we can see that the virus control and the virus flushed from the standard LDPE surface produced a high amount of S-protein, indicating that the translation of proteins occurred normally inside these cells. In contrast, the virus flushed from the rosin-functionalized plastic remained accumulated inside the vesicles (endosomes).

The percentage of infection in the standard LDPE-treated and rosin-functionalized plastic-treated viruses was normalized against the virus control. There were no apparent differences in the infectivity between the cells infected with the standard LDPE-treated surface and the virus control, as also seen with the CPE assay previously (Fig. 1B). However, the viruses treated on the rosin-functionalized surface were incapable of infecting the MRC-5 cells (Fig. 9B). Furthermore, we saw that the virus accumulated inside the vesicles did not proceed with replication, as no signal for dsRNA (green) was detected (Fig. 9C). These results indicated that the virus flushed from the rosin-function-alized plastic successfully binds onto the host cell, internalizes inside vesicles, but due to interactions with rosin compounds becomes incapable of proceeding with protein translation and viral replication.

## DISCUSSION

Respiratory viruses can spread through direct and indirect transmission routes. While various strategies are in place to limit direct transmission, such as travel restrictions, usage of face masks, and routine sanitization practices, there are fewer options available to address the indirect route of transmission through fomite surfaces. Repeated surface disinfection and the use of biocidal agents are commonly employed measures. The indirect route of virus transmission depends on several factors, most notably, the ability of the virus to remain active on the surface of a material. It is thus important to prepare for future outbreaks by designing antiviral surfaces that would effectively reduce the spread in the community. Functionalizing surfaces with biologically derived materials from nature to reduce indirect transmission is still relatively uncommon. However, due to their natural origin, such materials hold great promise as sustainable alternatives for microbial inactivation.

(A)

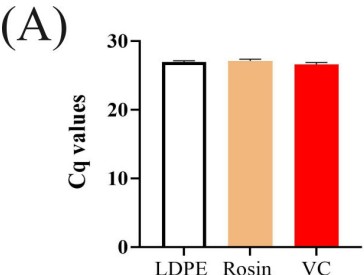

(B)

| DAPI | Tubulin | S protein | Merged |

*Virus Control*

*LDPE*

*Rosin*

**FIG 8** HCoV-OC43 virus binding onto its host cell receptors studied using (A) binding assay. The virus binding to host cell receptors was examined by facilitating synchronized virus binding onto the host cells, followed by recovery of the bound virus using RNA extraction, RT-PCR, and qPCR techniques. The experiment was performed three times with three technical replicates. VC, virus control. (B) Confocal microscopy. The viruses flushed from the surfaces were allowed to bind to MRC-5 cells for 1 h at RT and then internalize for another hour at 34°C. The virus-infected cells were fixed, permeabilized, and labeled with affinity-purified monoclonal rabbit anti-OC43-S protein, followed by a secondary goat anti-rabbit antibody (Alexa 555). The nucleus was stained blue with DAPI, and the cytoskeleton tubulin (gray) was labeled with a J2 primary antibody and goat anti-mouse secondary antibody (Alexa 488). Scale bar: 30 µm.

When studying antiviral surfaces, it is essential to first have a detailed understanding of the persistence of the virus on different fomite surfaces. Among the different known fomite surfaces, plastic demands particular attention. Although plastic is susceptible to degradation by disinfectants and UV-C, different grades of plastic have widespread use in household, commercial, and industrial settings. In the past, several studies have demonstrated good persistence of different human coronavirus species on plastic

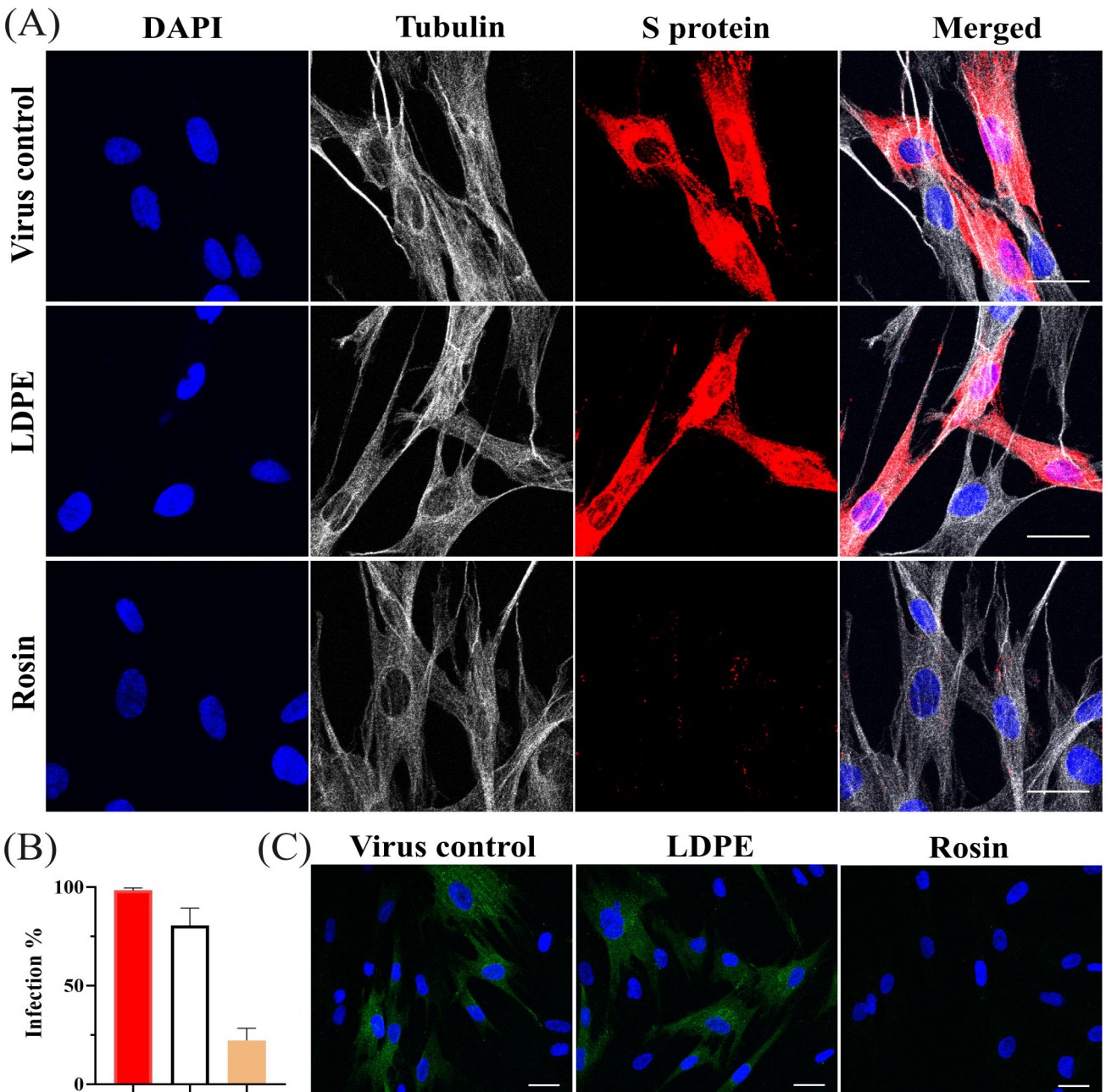

**FIG 9** (A) Confocal microscopy images illustrating the entrapment of the virus within vesicles 15 h after infection, following flushing from the rosin-functionalized surface. Viruses from both surfaces and the control virus were allowed to infect cells for 15 h, followed by fixing, permeabilizing, and immuno-labeling with antibodies. The S protein of the virus is labeled in red, the cell nucleus in blue, and the cell cytoskeleton tubulin in gray. (B) Percentage of virus-infected cells calculated from confocal images using Cell Profiler. (C) Detection of dsRNA in HCoV-OC43-infected MRC-5 cells 15 h p.i. The nuclei are shown in blue and the dsRNA in green. dsRNA is present in the virus control and the viruses flushed from the standard LDPE surface; however, the viruses flushed from the rosin-functionalized plastic do not show any dsRNA. Scale bar: 30 µm.

surfaces. For instance, Warnes and colleagues reported infective HCoV-229E on PTFE and polyvinyl chloride plastic surface for 5 days at 21°C and RH of 40%–50% (42). Independent studies by Duan et al. (43) and Rabenau et al. (44) reported the persistence of SARS-CoV (strain P9 and FFM1) on plastic at RT for up to 4 and 6 days, respectively (43, 44). Rabenau and colleagues reported further that the HCoV-229E could persist on plastic only for 24 h at RT. These variations in coronavirus persistence can be attributed to the interplay of multiple factors such as environmental conditions (temperature,

relative humidity, wind speed, etc.), type and surface topology of the plastic, virus type and concentration, and mode of deposition used in the experiments (11). While some publications do not specify the type of plastic being used in the study and generalize it as just "plastic," the type of polymers may affect the surface properties, such as hydrophobicity/hydrophilicity, which could also affect the persistence. While we lack standardized methods and conditions for testing, all the previous studies suggest that non-functionalized plastic retains the infectivity of coronaviruses well.

We focused here to explore the antiviral properties of a plastic surface functionalized with rosin (resin acid mixture) against human coronaviruses. HCoV-OC43 was chosen as a model for the study due to its structural and genetic similarity to the more virulent SARS-CoV-2 (45), allowing for work in a BSL-2 laboratory, although we also verified the results with SARS-CoV-2 as well. All the non-functionalized surfaces sustained high virus infectivity for the first 48 h, which was expected in the light of the previous studies (42–44). However, strikingly, the rosin-functionalized plastic exhibited remarkable antiviral efficacy against both the seasonal coronavirus and SARS-CoV-2 already within 15 min of contact time. Previously, several efforts have been made to add antiviral properties to surfaces, e.g., through functionalization with biocidal compounds (46), antiviral polymers (47, 48), metallic and nanoparticle coatings (49), and surface texturing (46, 50). In a study performed by Zhou and colleagues (50), an antiviral plastic surface designed by nanopatterning and coating of silver nanoparticles showed antiviral effect against SARS-CoV-2 after 1 h of contact time. In another recent study by Butot and colleagues (51), the antiviral activity of quaternary ammonium compounds (QACs) and copper-coated polyethylene terephthalate surface against HCoV-229E and SARS-CoV-2 was shown. Both surfaces were able to reduce viral titers, but the QAC-coated surface did not work after one round of washing. Here, we present a more sustainable solution using rosin-functionalized plastic. This surface showed an excellent 3–4 log reduction of SARS-CoV-2 and low toxicity against cells.

The survival time of the virus on a surface is influenced by various factors, among which RH plays a crucial role (11). RH can vary significantly across different seasons and indoor environments, impacting virus persistence. Previous studies have reported diverse patterns of virus persistence under different conditions of RH. For instance, Biryukov and colleagues (52) observed a linear decrease in SARS-CoV-2 persistence on an ABS plastic surface with increasing RH. In contrast, Morris and colleagues (53) found that the SARS-CoV-2 persisted for a longer period at extreme humidities, indicating a U-shaped relation with RH. However, we observed little effect of RH on the antiviral efficacy of the rosin-functionalized surface. The virus was inactivated at an earlier time point at lower humidities of 20% and 40% compared to higher humidity levels of 92%. However, rosin-functionalized plastic showed excellent inactivation within the first 30 min of contact with the functionalized surface in all tested humidities.

To gain a better understanding of the antiviral mechanism of the rosin-functionalized plastic, our study employed advanced imaging techniques such as AFM, TEM, HIM, and cryoEM to investigate the status and morphology of the flushed viruses. Interestingly, our investigations did not reveal any apparent structural changes in the virus, which could explain the observed loss of infectivity. This suggested that the rosin-functionalized surface had minimal impact on the size, shape, and physical dimensions of individual virus particles. Additionally, we investigated the structure of the virus on rosin-functionalized plastic using HIM. Notably, we observed distinct differences between the virus on the functionalized and the non-functionalized surfaces. On the functionalized surface, the viruses appear flatter in shape. This observation could be explained by the contact angle measurements, which suggested that the functionalized plastic was more hydrophilic (contact angle: 55°) compared to the non-functionalized one (contact angle: 95°). The hydrophilic nature of the functionalized plastic caused the droplets containing the virus to spread out, resulting in the flattening of the viruses. Despite this flattening, flushed viruses from rosin-functionalized surface did not appear different from the

viruses flushed from the non-functionalized plastic, suggesting that the hydrophilicity may play a minor role in killing the virus infectivity.

It is known that viruses tend to persist for longer periods on hydrophobic surfaces as opposed to hydrophilic surfaces (4). This behavior has been explained by Paton and colleagues (54), who proposed that on hydrophobic surfaces, droplets dry in a beaded shape, creating a higher volume-to-surface ratio. This drying process brings the virus particles, organic material, and salts in proximity, offering a protective environment against desiccation. Owen and colleagues (55) suggest further that faster desiccation from hydrophilic surfaces can potentially alter the pH and salt concentration inside the droplet, thereby affecting the stability of the lipids in the viral envelope. However, the hydrophilicity of the rosin-functionalized surface does not solely explain the observed antiviral effect. Namely, the quantification of viral RNA using qPCR demonstrated that the virus added to the surface was easily flushed off into the surrounding media, indicating a weak and temporary interaction between the virus and the hydrophilic surface.

Our imaging results as well as the fact that even short periods of encounter with the functionalized surface caused a strong antiviral effect, both pointed out to the direction that active rosin leaching out from the surface could be the cause of action. Indeed, we could demonstrate through absorption spectroscopy that the active ingredient inside the functionalized plastic leached out into the flush medium. Importantly, the leached flush was independently capable of causing the antiviral effect. Thus, it is likely that the leached rosin components contribute to most of the antiviral effect.

Previously, in addition to SARS-CoV-2, other enveloped viruses such as influenza A virus and respiratory syncytial virus have been shown to be affected by products derived from tall oil. However, non-enveloped viruses like encephalomyocarditis virus (EMCV) were not affected (20). Bell and colleagues, therefore, proposed that the lack of efficacy against non-enveloped viruses suggests the involvement of the lipids in the viral envelope by inhibiting the fusion between the viral envelope and host cell membranes. As bacteria also have membranes, it is no wonder that resin could have effects also on bacterial survival. Indeed, Sipponen and colleagues (56) showed the effect of Norway spruce-derived resin salves on bacterial membranes. The interaction of the resin salve caused the thickening of the bacterial membrane in *Staphylococcus aureus*. They additionally reported changes in the branching of the fatty acids and an altered membrane potential. Another study by Jokinen and Siponen (57) also demonstrated the antibacterial activity of refined spruce resin in treating chronic wounds. Tall oil rosin is a mixture of eight closely related resin acids, of which abietic acid and dehydroabietic acid have been more heavily studied for their antimicrobial properties. In a study by Agudelo-Gomez and colleagues (21), they not only showed antiherpetic activity (HHV-1 and HHV-2) of commercial abietic acid and dehydroabietic acid but also developed their analogs that displayed enhanced antiviral activity. Another study by Fonseca and colleagues (58) demonstrated the antiviral activity of dehydroabietic acid and their analogs against varicella-zoster virus and cytomegalovirus. We have preliminary results showing excellent antiviral efficacy of the rosin-functionalized plastic against the African Swine Fever Virus (J. D. Hemmink, S. Shroff, N. Chege, M. Haapakoski, L. K. Dixon, and V. Marjomäki, unpublished results). As this virus is known to be very difficult to decontaminate, these results suggest that rosin-functionalized plastic may also serve as an antiviral surface for more stable viruses and a larger variety of enveloped viruses. Furthermore, rosin chemicals are biobased, easy to apply to conventional polymer matrices, available in bulk, and very affordable, especially in comparison to several proposed nanomaterial applications. So, the industrial applications of this approach have promise in locations where surface contact plays a role in the transfer of pathogens like public spaces, hospitals, shops, transport, etc.

Using confocal microscopy, we show that, despite the 1 h encounter on the rosin surface with HCoV-OC43 and the resulting total block in infectivity, the viruses were successful in binding to MRC-5 cells. This indicates that the leached rosin components were not altering the functionality of the spike proteins to promote receptor binding.

This is in line with our cryoEM evaluation of the flushed viruses, which showed intact spikes that were mostly in the prefusion stage. As the binding to receptors was not inhibited, the virions were able to proceed and enter cytoplasmic vesicles. However, importantly, post entry, the viruses were stuck inside vesicles and did not proceed with replication and translation in the cytoplasm, which was demonstrated by confocal labeling of replication intermediates. It thus seems obvious that the block in infectivity was in the endosomal membrane fusion step that occurs between the viral envelope and the endosomal membrane. The two major sites on the virus surface involved in membrane fusion are the S2 subunit of the viral spike protein and the viral lipid-rich envelope (59). The S2 subunit has two heptad repeats HR1 and HR2 that form a six-helix bundle, which are responsible for membrane fusion and genome release. In addition to a possible direct effect on spike proteins and their processing, the viral membrane is likely to be affected due to interactions between the envelope lipids and the bioactive rosin components. This study was not able to pinpoint the mechanism of action by the leaching rosin; however, it did confirm its effective response in inactivating viruses on treated surfaces. It will be remarkably interesting to study and identify in the future what exact steps in the fusion process are affected by the rosin components.

Through this study, we demonstrate that the plastic surface functionalized with rosin effectively compromises the infectivity of coronaviruses. Importantly, the rosin-functionalized surface leaches enough rosin components to kill the fusion capability of coronaviruses when they have been in contact with the surface even for a relatively brief time. Due to the conserved nature of membrane fusion of enveloped viruses from different virus families, this rosin-functionalized plastic might serve as a broadly acting antiviral solution for a larger variety of viruses. Altogether, we propose that the rosin-functionalized plastic is a promising candidate as an antiviral surface for enveloped viruses.

## ACKNOWLEDGMENTS

We would like to thank Premix company for the plastic material for the study. A special acknowledgment to Petri Papponen from the University of Jyväskylä for his technical expertise in sample preparation and imaging with the HIM and TEM. We are grateful to Prof. Tarja Sironen (University of Helsinki) for letting us use the BSL-3 facility in Helsinki for testing our samples with SARS-CoV-2. We thank Moona Huttunen and Ilkka Julkunen from the University of Turku for providing the N-protein and S-protein antibodies against HCoV-OC43 for the confocal studies. We would like to thank Pasi Laurinmäki, Aušra Domanska, and Behnam Lak from the Biocenter Finland HiLIFE cryoEM Unit at the University of Helsinki within the Instruct-FI Centre for their assistance with CryoEM imaging and acknowledge the Instruct-ERIC (PID:23122; VID:39418) for providing the funding for the same.

This work was supported by a grant from Business Finland (4445/31/2021) to V.M., Academy of Finland grants to S.J.B. (336471 and 315950), the Sigrid Juselius Foundation (95-7202-38 to S.J.B.), and the Jane and Aatos Erkko Foundation (to S.J.B., V.M., and J.J.T.).

## AUTHOR AFFILIATIONS

[1]Department of Biological and Environmental Sciences, Nanoscience Center, University of Jyväskylä, Jyväskylä, Finland

[2]Department of Physics, Nanoscience Center, University of Jyväskylä, Jyväskylä, Finland

[3]Department of Neuroscience and Biomedical Engineering, Aalto University, Espoo, Finland

[4]Molecular and Integrative Bioscience Research Programme, Faculty of Biological and Environmental Sciences, University of Helsinki, Helsinki, Finland

[5]Institute of Biotechnology, Helsinki Institute of Life Sciences, University of Helsinki, Helsinki, Finland

[6]Sustainable Technologies group, Department of Chemistry, University of Eastern Finland, Joensuu, Finland

[7]FSCN Research Centre, Mid Sweden University, Sundsvall, Sweden

**AUTHOR ORCIDs**

Sarah J. Butcher  http://orcid.org/0000-0001-7060-5871
Varpu Marjomäki  http://orcid.org/0000-0002-4592-5926

**FUNDING**

| Funder | Grant(s) | Author(s) |
|---|---|---|
| Business Finland | 4445/31/2021 | Varpu Marjomäki |
| Academy of Finland (AKA) | 336471, 315950 | Sarah J. Butcher |
| Sigrid Juséliuksen Säätiö (Sigrid Jusélius Stiftelse) | 95-7202-38 | Sarah J. Butcher |
| Jane ja Aatos Erkon Säätiö (J&AE) | | Sarah J. Butcher |
| | | J. Jussi Topari |
| | | Varpu Marjomäki |

**AUTHOR CONTRIBUTIONS**

Sailee Shroff, Conceptualization, Data curation, Formal analysis, Investigation, Methodology, Writing – original draft | Marjo Haapakoski, Investigation, Writing – review and editing | Kosti Tapio, Conceptualization, Data curation, Formal analysis, Investigation, Methodology, Writing – original draft, Writing – review and editing | Mira Laajala, Investigation | Miika Leppänen, Conceptualization, Data curation, Formal analysis, Investigation, Methodology, Writing – original draft | Zlatka Plavec, Conceptualization, Data curation, Formal analysis, Investigation, Methodology, Writing – review and editing | Antti Haapala, Writing – original draft, Writing – review and editing | Sarah J. Butcher, Formal analysis, Writing – review and editing | Janne A. Ihalainen, Data curation, Formal analysis, Investigation, Methodology, Writing – review and editing | J. Jussi Topari, Conceptualization, Methodology, Writing – review and editing | Varpu Marjomäki, Conceptualization, Formal analysis, Funding acquisition, Methodology, Supervision, Writing – review and editing

**ADDITIONAL FILES**

The following material is available online.

Open Peer Review

**PEER REVIEW HISTORY (review-history.pdf).** An accounting of the reviewer comments and feedback.

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
