## [Reviewer comments · Microbiology Spectrum]

Microbiology Spectrum

Antiviral action of a functionalized plastic surface against human coronaviruses

Sailee Shroff, Haapakoski Marjo, Tapio Kosti, Mira Laajala, Leppänen Miika, Plavec Zlatka, Antti Haapala, Sarah Butcher, Janne Ihalainen, Toppari Jussi, and Varpu Marjomäki

Corresponding Author(s): Varpu Marjomäki, Jyväskylän yliopisto

Review Timeline:

Submission Date:	August 4, 2023
Editorial Decision:	October 11, 2023
Revision Received:	November 21, 2023
Editorial Decision:	November 28, 2023
Revision Received:	December 14, 2023
Accepted:	December 16, 2023

Editor: JJ Miranda

Reviewer(s): The reviewers have opted to remain anonymous.

Transaction Report:

DOI: <https://doi.org/10.1128/spectrum.03008-23>

October 11, 2023

Prof. Varpu Marjomäki
Jyvaskylan yliopisto
Biological and Environmental Sciences / Nanoscience center
Survontie 9
Jyväskylä, Jyväskylä 40500
Finland

Re: Spectrum03008-23 (Antiviral action of a functionalized plastic surface against human coronaviruses)

Dear Prof. Varpu Marjomäki:

Thank you for your contribution studying the effect of plastic surfaces on coronavirus structure and function using a wide variety of methods. I apologize for the length of time required for peer review. I only received your manuscript a few weeks ago as editor, but I can now provide feedback. The reviewers note overall strength in the breath of approaches to study coronaviruses on these surfaces, but they also have some questions regarding methodology. Addressing the comments provided would improve the manuscript.

Editorially, I will note what appears to be some confusion regarding interpretation of the qPCR experiments measuring SARS-CoV-2 on surfaces. Please revise the writing to avoid misinterpretation. More clearly distinguish between effects on infectivity as opposed to RNA destruction. My understanding is that the qPCR was used to measure the replication of recovered virus and not necessarily the amount of RNA on the surface itself.

Link Not Available

Sincerely,

JJ Miranda

Journals Department
Reviewer comments:

Reviewer #1 (Comments for the Author):

HCoV-OC43 is commonly used as a surrogate for SARS-CoV-2. It has been shown that the coronaviruses do not have always the same stability on surfaces, even if decontaminants can have identical effect on both viruses. Therefore, it is disappointing that the authors did not validate the results on SARS-CoV-2 using the same protocols and techniques to compare the effect of rosin-functionalized plastic the way they did for HCoV-OC43, just because of the biosafety level of the viruses. It would be interesting to do at least the Quantification of HCoV-OC43 viral RNA, using the same protocols as for SARS-CoV-2. Indeed, the effect of rosin-functionalized plastic is significant has shown by figure 1 (CPE assay, 1A & 1B), whereas the amount of SARS-CoV-2 RNA destroyed on the rosin-functionalized plastic compared to the standard LDPE is observed. It is difficult to interpreted since it is shown in this work that no apparent structural changes in the HCoV-OC43 virus could be observed meaning that the viruses are not destroyed (1C). The manuscript would be improved if the same qPCR assays could be done on HCoV-OC43 (to improve/comfort the results in absence of CPE assays for SARS-CoV-2) and to see if HCoV-OC43 RNA are also destroyed. It would also strengthen the conclusion on figure 3, if the authors can demonstrate that HCoV-OC43 RNA are destroyed by using the same conditions as 2.5 (with SARS-CoV-2).

Reviewer #2 (Comments for the Author):

Summary

The study investigated the effectiveness of plastic surfaces functionalized with tall-oil rosin in inactivating human coronaviruses, specifically HCoV-OC43 and SARS-CoV-2. Here are the key results from the study:

Antiviral Activity: The research found that plastic surfaces treated with tall-oil rosin demonstrated significant antiviral activity against both HCoV-OC43 and SARS-CoV-2. This indicates that the rosin-functionalized plastic can effectively reduce the infectivity of these coronaviruses on surfaces.

Inactivation Over Time: The study examined the persistence of these viruses on the rosin-functionalized plastic surfaces over various time intervals. The results showed a time-dependent reduction in viral infectivity, with longer exposure leading to greater inactivation. This suggests that the rosin-coated plastic becomes more effective at neutralizing the viruses with extended contact time.

Comparison to Standard Plastic: The research compared the antiviral properties of the rosin-functionalized plastic with standard low-density polyethylene (LDPE) plastic. The rosin-coated plastic consistently outperformed the standard plastic in reducing viral infectivity, indicating its superiority in preventing viral transmission.

Effect of Humidity: The study also assessed the impact of humidity levels on the antiviral function of the rosin-functionalized plastic. It found that the antiviral effect remained effective even under different humidity conditions, making it a versatile solution for various environmental settings.

Quantification of Viral RNA: The researchers quantified the viral RNA flushed from the surfaces using PCR (polymerase chain reaction) analysis. This allowed them to calculate the amount of viral RNA destroyed on the rosin-functionalized plastic compared to standard plastic. The results showed a significant reduction in viral RNA on the rosin-treated surfaces, indicating effective viral inactivation.

Major Concerns

- My only large concern is that although I'm convinced there is antiviral activity due to the 10% rosin LDPE, I hesitate to overstate the impact of these findings. Most notably, coronaviruses are enveloped viruses which are among the easiest pathogens to kill via disinfectants, so although we see an impact: 1) is it a strong enough signal to convince us that we should pursue more difficult pathogens next and 2) is something of this nature cost-effective if it were to only work on pathogens low on the disinfectant pedagogy. I would suggest the authors take this into the discussion section and minimally list it as a study limitation. Although my concern seems pessimistic, I absolutely believe we need continuous or novel disinfectant strategies to combat the human component of disinfection and cleaning, just like this potential one!

Minor Concerns

- A few minor notes to make note of in the limitations section of the paper
- The use of a glass coverslip to maximize contact likely enhanced the disinfection properties of all surfaces and removed the environment from the experiment, I would note this.
- The use of CTs even in a controlled manor is always a risk and limitation as they are semi-quantitative at best.
- The authors do not know the antiviral action or method of the rosin and as such all experiments should be taken with that in mind (does it attack genetic material, membrane, etc. impacts different potential assays)
- Also, the use of RNA as a marker should be listed as a limitation due to natural degradation outside of the host virus and unknown rosin antiviral activity.
- The experiments were over a short period of time - do the authors have data beyond these time points or literature/reason to stop at these points?
- Lastly, the authors should speak to potential downsides of the rosin LDPE: cost, susceptible to degradation via disinfectants, weak to UV-C (as most plastics are) to put it into context even if it does prove to work against other pathogens.

Notes

1. Lines 22-29 and 30-37 are the same text repeated.
2. Section 2.5:
 - a. Were inoculated surfaces all flushed at time intervals? Or were unique surfaces only flushed at a single time interval? If a single surface was flushed several times, this would cause a decrease in viral load each sampling and create a false positive decline of virus.

Staff Comments:

Preparing Revision Guidelines

Please return the manuscript within 60 days; if you cannot complete the modification within this time period, please contact me. If you do not wish to modify the manuscript and prefer to submit it to another journal, please notify me of your decision immediately so that the manuscript may be formally withdrawn from consideration by Microbiology Spectrum.

Review of Antiviral plastic surface against human coronaviruses

Comments and Suggestions for the Author:

- In the introduction lanes 22-23 and 30-31 are identical except the link to ref (8-12 and 5-8). Please modify.
- It is the same for lanes 24-29 and 32-37. Please revised the introduction and you should verify that the links to the ref are correct.
- *A virus without any surface treatment was used as a positive control and MRC-5*

116 cells without any virus infection were used as a negative control (or mock infection).

117 Additionally, a sample control without the addition of the virus was used to test the cytotoxicity

118 of the surfaces on MRC-5 cells.

In this part (Materials and methods), the difference between the MRC-5 cells used as negative control and the following lanes 117-118 is not clear (again MRC-5 cells without addition of the virus). Is it the same conditions in the 2 sentences or in the first case, the culture is made in the well plate and in the second case, culture in well plates and then deposited on PREXELENT?

- Lane 181 : should had in the text that the virus used is HCoV-OC43
- Lane 206 : should had in the text that the virus used is HCoV-OC43
- Lane 219 : should had in the text that the virus used is HCoV-OC43
- Lane 238 : should had in the text that the virus used is HCoV-OC43
- Lane 257 : should had in the text that the virus used is HCoV-OC43
- Lane 275 : should had in the text that the virus used is HCoV-OC43
- Lane 289 : should had in the text that the virus used is HCoV-OC43
- Lane 300 : should had in the text that the virus used is HCoV-OC43
- Lane 454 : should had in the text that the virus used is HCoV-OC43
- Lane 492 : should had in the text that the virus used is HCoV-OC43
- Lane 871 : should had in the text that the RNA used comes from HCoV-OC43
- Lane 884 : should had in the text that the virus used is HCoV-OC43
- Lane 893 : should had in the text that the virus used is HCoV-OC43

Reviewer 1

HCoV-OC43 is commonly used as a surrogate for SARS-CoV-2. It has been shown that the coronaviruses do not always have the same stability on surfaces, even if decontaminants can have identical effect on both viruses. Therefore, it is disappointing that the authors did not validate the results on SARS-CoV-2 using the same protocols and techniques to compare the effect of rosin-functionalized plastic the way they did for HCoV-OC43, just because of the biosafety level of the viruses. It would be interesting to do at least the Quantification of HCoV-OC43 viral RNA, using the same protocols as for SARS-CoV-2.

Answer: We agree with the reviewer that it would be most optimal if all experiments were done with SARS-CoV-2. But as the referee points out, the need for higher biosafety level has made the work more complicated. As suggested by the referee, we have now performed the quantification of HCoV-OC43 viral RNA, using the same protocol as for SARS-CoV-2. The results of the experiment are now illustrated in Figure below (now also added to modified Figure 1C). We tested the same time points for HCoV-OC43 as what was performed for SARS-CoV-2 (but added also additional long time point for HCoV-OC43 as suggested by the second reviewer). The results of the qPCR from the RNA collected 3 days post infection (the same procedure was done for SARS-CoV-2 as well) showed that ample infection (high amount of viral RNA measured from 3 day cultivation in cells) could be detected for all the timepoints on the LDPE surface. In contrast, the amount of viral RNA dropped rapidly after 5 minutes suggesting that the rosin surface was very effective for the seasonal coronaviruses, like with SARS-CoV-2, measured with similar method (line number 422-443).

Figure: Persistence of HCoV-OC43 on standard LDPE and rosin-functionalized plastic measured up to 24 h of contact time using qPCR.

Indeed, the effect of rosin-functionalized plastic is significant has shown by figure 1 (CPE assay, 1A & 1B), whereas the amount of SARS-CoV-2 RNA destroyed on the rosin-functionalized plastic compared to the standard LDPE is observed. It is difficult to interpreted since it is shown in this work that no apparent structural changes in the HCoV-OC43 virus could be observed meaning that the viruses are not destroyed (1C). The manuscript would be improved if the same qPCR assays could be done on HCoV-OC43 (to improve/comfort the results in absence of CPE assays for SARS-CoV-2) and to see if HCoV-OC43 RNA are also destroyed. It would also strengthen the conclusion on figure 3, if the authors can demonstrate that HCoV-OC43 RNA are destroyed by using the same conditions as 2.5 (with SARS-CoV-2).

Answer: We are sorry for the confusion of the outcome of the qPCR methods and comparison to CPE method. We have now tried to add clarity and simplified the methods section. Also, we have changed our wording about the results. We do not talk about “destroyed RNA” but rather the amount of infectious virus calculated by the RNA content of the viruses produced in the culture supernatant (done

in Figure 1), vs. viral RNA measured directly from the media after flushing of the surface (Figure 3). So, the presence of viral RNA in the Figure 3 has been performed in a different manner from the Figure 1C, although also with the same qPCR technique. In the Figure 1C we have measured infection using the qPCR to quantitate the relative amount of RNA (virus) produced in cells after 3 day cultivation. This gives us a possibility to measure infection in a different manner than the CPE which measures cell viability (Figures. 1A, B). In contrast, in the Figure 3 we wanted to measure how much of the bound virus was directly flushed away from the surface. Therefore, in Figure 3, we quantitated the amount of RNA from the flushed media, and not after 3 day cultivation in cells (line number 422-443 and 478-486).

In the introduction lanes 22-23 and 30-31 are identical except the link to ref (8-12 and 5-8). Please modify. It is the same for lanes 24-29 and 32-37. Please revised the introduction and you should verify that the links to the ref are correct.

Answer: Thank you for noticing this. We have now deleted excess lines and checked the citation.

A virus without any surface treatment was used as a positive control and MRC-5 cells without any virus infection were used as a negative control (or mock infection). Additionally, a sample control without the addition of the virus was used to test the cytotoxicity of the surfaces on MRC-5 cells. In this part (Materials and Methods), the difference between the MRC-5 used as a negative control and the following lanes 117-118 is not clear (again MRC-5 cells without addition of the virus). Is it the same condition in the 2 sentences or in the first case, the culture is made in the well plate and in the second case , culture in well plates and then deposited on PREXELENT?

Answer: We apologize that the description of the controls was not clear. We have added sentences in the material and methods section to clarify this (section 2.4): We used several controls in the experiment: 1) An equivalent amount of virus without any surface treatment was used as a positive control and 2) MRC-5 cells without any virus infection were used as a negative control (or mock infection). Additionally, 3) a sample control without the addition of the virus was used to test the cytotoxicity of the surfaces on MRC-5 cells (line number 147-151).

Lane 181 : should had in the text that the virus used is HCoV-OC43

Answer: HCoV-OC43 added to the text instead of virus

Lane 206 : should had in the text that the virus used is HCoV-OC43

Answer: HCoV-OC43 added to the text

Lane 219 : should had in the text that the virus used is HCoV-OC43

Answer: HCoV-OC43 added to the text

Lane 238 : should had in the text that the virus used is HCoV-OC43

Answer: HCoV-OC43 added to the text

Lane 257 : should had in the text that the virus used is HCoV-OC43

Answer: HCoV-OC43 added to the text

Lane 275 : should had in the text that the virus used is HCoV-OC43

Answer: HCoV-OC43 added to the text

Lane 289 : should had in the text that the virus used is HCoV-OC43

Answer: HCoV-OC43 added to the text

Lane 300 : should had in the text that the virus used is HCoV-OC43

Answer: HCoV-OC43 added to the text

Lane 454 : should had in the text that the virus used is HCoV-OC43

Answer: HCoV-OC43 added to the text

Lane 492 : should had in the text that the virus used is HCoV-OC43

Answer: HCoV-OC43 added to the text

Lane 871 : should had in the text that the RNA used comes from HCoV-OC43

Answer: HCoV-OC43 added to the text

Lane 884 : should had in the text that the virus used is HCoV-OC43

Answer: HCoV-OC43 added to the text

Lane 893 : should had in the text that the virus used is HCoV-OC43

Answer: HCoV-OC43 added to the text

Reviewer 2

Summary

The study investigated the effectiveness of plastic surfaces functionalized with tall-oil rosin in inactivating human coronaviruses, specifically HCoV-OC43 and SARS-CoV-2. Here are the key results from the study:

Antiviral Activity: The research found that plastic surfaces treated with tall-oil rosin demonstrated significant antiviral activity against both HCoV-OC43 and SARS-CoV-2. This indicates that the rosin-functionalized plastic can effectively reduce the infectivity of these coronaviruses on surfaces.

Inactivation Over Time: The study examined the persistence of these viruses on the rosin-functionalized plastic surfaces over various time intervals. The results showed a time-dependent reduction in viral infectivity, with longer exposure leading to greater inactivation. This suggests that the rosin-coated plastic becomes more effective at neutralizing the viruses with extended contact time.

Comparison to Standard Plastic: The research compared the antiviral properties of the rosin-functionalized plastic with standard low-density polyethylene (LDPE) plastic. The rosin-coated plastic consistently outperformed the standard plastic in reducing viral infectivity, indicating its superiority in preventing viral transmission.

Effect of Humidity: The study also assessed the impact of humidity levels on the antiviral function of the rosin-functionalized plastic. It found that the antiviral effect remained effective even under different humidity conditions, making it a versatile solution for various environmental settings.

Quantification of Viral RNA: The researchers quantified the viral RNA flushed from the surfaces using PCR (polymerase chain reaction) analysis. This allowed them to calculate the amount of viral RNA destroyed on the rosin-functionalized plastic compared to standard plastic. The results showed a significant reduction in viral RNA on the rosin-treated surfaces, indicating effective viral inactivation.

Major Concerns

My only large concern is that although I'm convinced there is antiviral activity due to the 10% rosin LDPE, I hesitate to overstate the impact of these findings. Most notably, coronaviruses are enveloped viruses which are among the easiest pathogens to kill via disinfectants, so although we see an impact:

1) is it a strong enough signal to convince us that we should pursue more difficult pathogens next

Answer: We do agree with the referee that the infectivity of enveloped viruses is easier to kill than e.g., that of non-enveloped viruses. However, as we observed with the dense and uniform plastic surfaces, the infectivity of coronaviruses stays high even after 2 days, subject to factors like temperature and humidity. Therefore, we think that to lower the infectivity in fomites without cleaning or disinfection activities, this rosin-embedded surface serves as an excellent option. In addition, we have now performed experiments on viruses that although being enveloped, are known to be very stubborn and which are a real threat worldwide: we have results that demonstrate that the same rosin surface can kill infectivity of the African Swine Fever virus while the hard surfaces without functionalization do not kill infectivity (J.D.Hemmink, S. Shroff, N. Chege, M. Haapakoski, L.Dixon. V. Marjomaki; manuscript in preparation). We are excited about the possibility to kill the infectivity of this very stable type of viruses, and it gives promise of this surface to act on a larger variety of enveloped viruses (line number 726-730).

2) is something of this nature cost-effective if it were to only work on pathogens low on the disinfectant pedagogy. I would suggest the authors take this into the discussion section and minimally list it as a study limitation. Although my concern seems pessimistic, I absolutely believe we need continuous or novel disinfectant strategies to combat the human component of disinfection and cleaning, just like this potential one!

Answer: As stated above, we believe that the rosin-surface has potential to kill a larger variety of pathogens and the rosin chemicals applied are biobased, easy to apply to conventional polymer matrices, available in bulk and very affordable (especially in comparison to several proposed nanomaterial applications), so the industrial applications of this approach have promise in locations where surface contact plays a role in the transfer of pathogens (like public spaces, hospitals, shops, transport, etc.). This has been added to the discussion section (line number 730-734).

Minor Concerns

A few minor notes to make note of in the limitations section of the paper

The use of a glass coverslip to maximize contact likely enhanced the disinfection properties of all surfaces and removed the environment from the experiment, I would note this.

Answer: It is great that the referee pointed out this point concerning the use of the coverslip. We had forgotten to mention in the material and methods that, for specified humidities tested in the custom chamber, we did not use coverslip on top of the plastic and virus. We have now added this in the material and methods (at the end of section 2.4) (line number 176-178).

The use of CTs even in a controlled manor is always a risk and limitation as they are semi-quantitative at best.

Answer: The referee comment is justified. We have modified and simplified the explanation of the qPCR part in the material and methods section. Also, we have mentioned that with the method we are comparing the relative amounts of viral RNA between the samples (section 2.5) (line number 205-209).

The authors do not know the antiviral action or method of the rosin and as such all experiments should be taken with that in mind (does is attack genetic material, membrane, etc. impacts different potential assays)

Answer: We agree with the referee. We have stated in the discussion section that we could not pinpoint the mechanism of action although the results point to compromised fusion. However, we are very interested to study this further (line number 750-752).

Also, the use of RNA as a marker should be listed as a limitation due to natural degradation outside of the host virus and unknown rosin antiviral activity.

Answer: We have now stated that the viral RNA produced in cells after 3 day of virus multiplication is used to compare the relative amounts of RNA with qPCR. When we measured the viral RNA directly after flushing of the surfaces (to measure how much of the input virus was stuck on the surface), we immediately isolated the RNA to secure its stability. Indeed, in the measurement, approximately the same amount of the RNA was measured in comparison to the input virus amount suggesting that, in those experiments, the RNA was not degraded before measuring.

The experiments were over a short period of time - do the authors have data beyond these time points or literature/reason to stop at these points?

Answer: This is a very good point by the referee. Indeed, we now also performed one longer time point, 24 h, both by CPE and qPCR for HCoV-OC43. These results have now been added in the modified Figure 1B (CPE) and 1C (qPCR). The results show clearly that the infectivity on LPDE is still very good after 24 h but totally lost in functionalized plastic by both methods. The same type of study has been tested repeatedly in several times in parallel projects (not reported here) with comparable results on diverse set of solid materials, so we feel confident that the methodology provides accurate and repeatable results also in the contact studies of quite brief timeframe (line number 394-421).

Lastly, the authors should speak to potential downsides of the rosin LDPE: cost, susceptible to degradation via disinfectants, weak to UV-C (as most plastics are) to put it into context even if it does prove to work against other pathogens.

Answer: We have now added these characteristics to the discussion. However, we feel that as the rosin plastic does not need actual disinfection as it disinfects itself, it can be cleaned with less degradative solutions (line number 625-626).

Notes

Lines 22-29 and 30-37 are the same text repeated.

Answer: We are sorry for this mistake. We have now omitted the duplicated text.

Section 2.5:

Were inoculated surfaces all flushed at time intervals? Or were unique surfaces only flushed at a single time interval? If a single surface was flushed several times, this would cause a decrease in viral load each sampling and create a false positive decline of virus.

Answer: Unique surfaces were flushed at a single time interval.

Re: Spectrum03008-23R1 (Antiviral action of a functionalized plastic surface against human coronaviruses)

Dear Prof. Varpu Marjomäki:

Thank you for the privilege of reviewing your work. Below you will find my comments, instructions from the Spectrum editorial office, and the reviewer comments.

Thank you for your revised submission that addressed the concerns of the reviewers thoroughly. As you are aware, Spectrum emphasizes rigor and reproducibility. Please make one more important modification to the manuscript text. While statistical tests and error bars are clearly indicated in the figures, sample size information is needed to communicate reproducibility. For each experiment, please indicate the number of times the experiment was reproduced. Clearly distinguish between technical and biological replicates, especially with regard to qPCR experiments.

I want to emphasize that only changes in text are requested at this stage. No experiments are necessary. I look forward to the final version of the manuscript.

Revision Guidelines

Sincerely,
JJ Miranda
Editor
Microbiology Spectrum

Re: Spectrum03008-23R1 (Antiviral action of a functionalized plastic surface against human coronaviruses)

Dear Prof. Varpu Marjomäki:

Thank you for the privilege of reviewing your work. Below you will find my comments, instructions from the Spectrum editorial office, and the reviewer comments.

Thank you for your revised submission that addressed the concerns of the reviewers thoroughly. As you are aware, Spectrum emphasizes rigor and reproducibility. Please make one more important modification to the manuscript text. While statistical tests and error bars are clearly indicated in the figures, sample size information is needed to communicate reproducibility. For each experiment, please indicate the number of times the experiment was reproduced. Clearly distinguish between technical and biological replicates, especially with regard to qPCR experiments.

I want to emphasize that only changes in text are requested at this stage. No experiments are necessary. I look forward to the final version of the manuscript.

Dear Editor Miranda,

Thank you for your kind words. We have now edited the text in the figure legends, and added notion of the number of experiments and technical repeats. We have indicated the changes in red color in a marked-up manuscript.

We hope that the manuscript is now acceptable for Microbiology Spectrum.

Re: Spectrum03008-23R2 (Antiviral action of a functionalized plastic surface against human coronaviruses)

Dear Prof. Varpu Marjomäki:

Thank you for your thorough and multidisciplinary study contributing new understanding toward the development of novel antiviral surfaces!

Your manuscript has been accepted, and I am forwarding it to the ASM production staff for publication. Your paper will first be checked to make sure all elements meet the technical requirements. ASM staff will contact you if anything needs to be revised before copyediting and production can begin. Otherwise, you will be notified when your proofs are ready to be viewed.

Sincerely,
JJ Miranda
Editor
Microbiology Spectrum